# The vaccinia virus DNA polymerase structure provides insights into the mode of processivity factor binding

Nicolas Tarbouriech[1], Corinne Ducournau[2], Stephanie Hutin[1], Philippe J. Mas[3], Petr Man[4,5], Eric Forest[1], Darren J. Hart[1], Christophe N. Peyrefitte[2,6], Wim P. Burmeister[1] & Frédéric Iseni[2]

Vaccinia virus (VACV), the prototype member of the *Poxviridae*, replicates in the cytoplasm of an infected cell. The catalytic subunit of the DNA polymerase E9 binds the heterodimeric processivity factor A20/D4 to form the functional polymerase holoenzyme. Here we present the crystal structure of full-length E9 at 2.7 Å resolution that permits identification of important poxvirus-specific structural insertions. One insertion in the palm domain interacts with C-terminal residues of A20 and thus serves as the processivity factor-binding site. This is in strong contrast to all other family B polymerases that bind their co-factors at the C terminus of the thumb domain. The VACV E9 structure also permits rationalization of polymerase inhibitor resistance mutations when compared with the closely related eukaryotic polymerase delta–DNA complex.

[1] Institut de Biologie Structurale (IBS), Université Grenoble Alpes, CNRS, CEA, 71 Avenue des Martyrs, 38042 Grenoble, France. [2] Unité de Virologie, Institut de Recherche Biomédicale des Armées, BP 73, 91223 Brétigny-sur-Orge Cedex, France. [3] Integrated Structural Biology Grenoble (ISBG) CNRS, CEA, Université Grenoble Alpes, EMBL, 71 Avenue des Martyrs, 38042 Grenoble, France. [4] BioCeV-Institute of Microbiology, Czech Academy of Sciences, Prumyslova 595, 252 50 Vestec, Czech Republic. [5] Faculty of Science, Charles University, Hlavova 8, 128 43 Prague 2, Czech Republic. [6] Emerging Pathogens Laboratory, Fondation Mérieux, 21 Avenue Tony Garnier, 69007 Lyon, France. Eric Forest Deceased. Correspondence and requests for materials should be addressed to F.I. (email: fredericiseni@gmail.com)

Poxviruses (members of the *Poxviridae* family) are large double-stranded DNA viruses that replicate exclusively in the cytoplasm of the infected cell. Viral DNA synthesis takes place in perinuclear sites called viral factories and depends on virus-encoded proteins[1]. Poxviruses produce a number of non-essential enzymes involved in DNA precursor metabolism as well as essential proteins that are located at the replication fork. For vaccinia virus (VACV), the prototype and most-studied member of the *Poxviridae* family, the essential replication proteins include: E9, the catalytic subunit of the DNA polymerase[2]; D4, a uracil-DNA glycosylase[3,] which together with A20 forms the heterodimeric processivity factor[4]; D5, a hexameric nucleoside triphosphatase[5,6], which contains a superfamily III helicase domain[7] and shows primase activity[8] and I3, a single-stranded DNA-binding protein[9]. Other members of the virally encoded replication machinery include G5, a FEN-family endonuclease; A50, a DNA ligase and H5, an abundant hub protein[10]. A low-resolution model of the VACV DNA polymerase holoenzyme E9/A20/D4 highlighted the elongated shape of the complex with a 150 Å separation between the DNA-binding sites of E9 and D4[11]. A20 links both enzymes, and the DNA-binding properties of D4 are believed to increase the association of E9 with the genome template thus rendering the polymerase processive[12]. Recently, high-resolution structures of the D4/A20 interface (D4/A20$_{1-50}$) and of D4/A20$_{1-50}$ bound to a 10-mer DNA duplex containing an abasic site resulting from the cleavage of an uracil base were obtained[13,14]. These data further extend our knowledge on the processivity factor assembly and how DNA synthesis and base excision repair are coupled. However, structural information concerning the DNA polymerase and its interaction with A20 is still missing.

Over the years, a number of genetic and biochemical studies have characterized E9 (reviewed by Czernecky and Traktman[10]). The enzyme is a member of the DNA polymerase family B[15] possessing DNA polymerase and 3′−5′ proofreading exonuclease activities[2]. E9 alone was shown to be distributive under physiological conditions (adding fewer than 10 nucleotides per binding event[16]) unless bound to its heterodimeric cofactor D4/A20[12].

**Table 1 Data collection and refinement statistics**

| | E9-WT native | E9-WT Pb$^{2+}$ | E9-WT Gd$^{3+}$ | E9-WT Mn$^{2+}$ | E9-exo$^{minus}$ |
|---|---|---|---|---|---|
| *Data collection* | | | | | |
| Beamline | ESRF ID23-1 | ESRF BM14 | ESRF BM14 | ESRF ID23-1 | ESRF ID23-1 |
| Space group | $P3_121$ | $P3_121$ | $P3_121$ | $P3_121$ | $P3_121$ |
| Cell parameters (Å) | 133.4 133.4 230.5 | 133.7 133.7 229.6 | 133.6 133.6 229.8 | 134.0 134.0 230.2 | 133.5 133.5 229.5 |
| Wavelength (Å) | 0.9762 | 0.9464 | 1.4226 | 1.0714 | 1.2724 |
| Resolution range (Å)$^a$ | 46.2-2.74 (2.81-2.74) | 46.2-3.79 (4.10-3.79) | 46.2–3.1 (3.22–3.10) | 46.3-2.79 (2.89-2.79) | 57.0-2.81 (2.96-2.81) |
| No. of observed reflections$^a$ | 332899 (22308) | 176231 (35375) | 236574 (24674) | 199915 (19915) | 188465 (26270) |
| No. of unique reflections$^a$ | 63014 (4369) | 23925 (4779) | 43418 (4429) | 59597 (5772) | 57789 (8176) |
| Completeness (%)$^a$ | 99.9 (99.9) | 99.4 (98.1) | 99.5 (98.9) | 99.1 (99.2) | 99.0 (97.1) |
| Multiplicity$^a$ | 5.3 (5.1) | 7.4 (7.4) | 5.4 (5.6) | 3.4 (3.5) | 3.3 (3.2) |
| Mean $I/\sigma(I)$$^a$ | 18.2 (2.8) | 15.9 (5.6) | 17.0 (5.2) | 11.6 (1.8) | 11.3 (1.8) |
| $R_{r.i.m}$$^{a,b}$ | 0.077 (0.728) | 0.141 (0.473) | 0.115 (0.451) | 0.108 (0.958) | 0.089 (0.788) |
| $R_{sym}$ (%) | 6.9 (65.3) | 13.1 (44.0) | 10.4 (40.5) | 9.1 (81.0) | 7.4 (65.3) |
| $CC_{1/2}$ | 0.999 (0.862) | 0.997 (0.944) | 0.996 (0.942) | 0.996 (0.679) | 0.996 (0.631) |
| Mosaicity (°) | 0.07 | 0.12 | 0.06 | 0.03 | 0.06 |
| Overall $B$ factor (Wilson plot) (Å$^2$) | 49.0 | 66.1 | 45.6 | 58.8 | 43.8 |
| *Model refinement and composition* | | | | | |
| No. of reflections, working set | | 59939 | | 56686 | 54984 |
| No. of reflections, test set | | 3075 | | 2911 | 2805 |
| Final $R_{cryst}$ | | 0.186 | | 0.183 | 0.185 |
| Final $R_{free}$ | | 0.236 | | 0.227 | 0.222 |
| No. of non-H atoms | | | | | |
| Protein | | 8188 | | 8180 | 8173 |
| Ligand | | 71 | | 75 | 71 |
| Water | | 219 | | 215 | 214 |
| Total | | 8478 | | 8470 | 8458 |
| *Model composition* | | | | | |
| Protein (residues) | | 999 | | 999 | 999 |
| MES | | 3 | | 3 | 3 |
| HEPES | | 1 | | 1 | 1 |
| DTT | | 1 | | 1 | 1 |
| Glycerol | | 2 | | 2 | 2 |
| Mn$^{2+}$ | | 0 | | 4 | 0 |
| *RMS deviations* | | | | | |
| Bond lengths (Å) | | 0.015 | | 0.011 | 0.011 |
| Bond angles (°) | | 1.855 | | 1.553 | 1.617 |
| Average $B$ factor (Å$^2$) | | 74 | | 72 | 78 |
| Ramachandran plot (%) | | | | | |
| Favored | | 92.4 | | 94.3 | 93.1 |
| Additionally allowed | | 6.3 | | 4.5 | 5.8 |
| Outliers | | 1.3 | | 1.2 | 1.1 |

$^a$Values in parentheses correspond to the highest-resolution shell
$^b$Estimated $R_{r.i.m} = R_{sym} \times [N/(N-1)]^{1/2}$, where $N$ is the data multiplicity

The DNA polymerase was also shown to catalyze annealing of single-stranded DNA[17], an activity not found in other DNA polymerase family B members. The end-joining reaction requires the 3′–5′exonuclease activity of E9 that degrades the extremities of dsDNA to create 5′-ssDNA overhangs[18]. Sequence alignments with other DNA polymerases identified E9-specific insertions but they have not yet been correlated with precise functions[11].

E9 is the target of several inhibitors such as aphidicolin (aph), phosphonoacetic acid (PAA), cytosine arabinoside (AraC), and CMX001, a cidofovir (CDV) derivative, which is in advanced development for the treatment of smallpox[19]. These compounds have been used to select and characterize resistance mutations located in the VACV DNA polymerase[20–27].

Here we present the 2.7 Å crystal structure of the full-length VACV DNA polymerase and the characterization of the interface between E9 and its processivity factor subunit A20. The data allow us to explore the role of E9-specific inserts and to position E9 in a global model of the DNA polymerase holoenzyme which differs from other family B polymerases.

## Results

**Crystal structure of E9 the VACV DNA polymerase**. E9 crystallized in space group $P3_121$ with one molecule in the asymmetric unit. The structure was solved at 2.7 Å resolution using the MIRAS method (Table 1). We observed the classical palm, thumb, finger, exonuclease, and N-terminal domains of a family B polymerase in an open conformation (Fig. 1a). Only 10 out of the 1006 residues could not be modeled. A previous sequence analysis of E9 allowed the delimitation of "poxvirus-specific" inserts[11] that we can now redefine based on flexible structural alignments with other family B polymerases. Three poxvirus-specific inserts are clustered on one side of the molecule, corresponding to insert 0 (aa 67–82), insert 3 (aa 572–610), and insert 4 (aa 708–743) (Fig. 1b), whereas insert 1 (aa 208–233) is located on the opposite side (Fig. 1c). Insert 2 (aa 356–432) is located on the back of E9 in the exonuclease domain (Fig. 1c) and forms a little 6-stranded β-barrel with a greek key fold. The previously assigned insert 5 does not exist whilst a new insert in the N-terminal domain is identified and denominated as insert 0 (Fig. 1b).

The crystal packing involves mainly the periphery of the molecule, implicating N-terminal, exonuclease, palm, and thumb domains. The helix of insert 3 forms an important contact as it is bound in a hydrophobic cavity of a symmetry-related molecule mainly comprising residues of the exonuclease domain (i.e., residues 190–200 and 227–233). The temperature factors indicate a high mobility of insert 2 (which appears to be in loose contact with the body of the polymerase) and of the thumb domain. A high mobility of the thumb domain is generally observed for family B polymerase. Still, for E9 electron density is visible for all the C-terminal residues.

A calculated electrostatic potential of E9 shows extensive positively charged surfaces on both faces of the molecule (Fig. 1d). On the front face (defined by the inferred location of the polymerase active site), these coincide with areas of conserved sequence within orthopoxviruses (Fig. 1e). On the back, however, little sequence conservation is observed.

**Identification of a soluble A20 domain interacting with E9.** Within the VACV DNA polymerase holoenzyme, A20 forms a link between D4 and the catalytic subunit of the polymerase E9[11,12]. While the first 50 residues of A20 were shown to interact with D4[14,28], the low-resolution structure of E9/A20/D4 suggests that the A20 C-terminal region may be involved in E9 interaction[11]. We used the ESPRIT technology[29] to identify soluble

purifiable C-terminal fragments of A20. A random library of 5′ truncations of the full-length *A20R* gene was generated using exonuclease III and mung bean nuclease. Approximately 4600

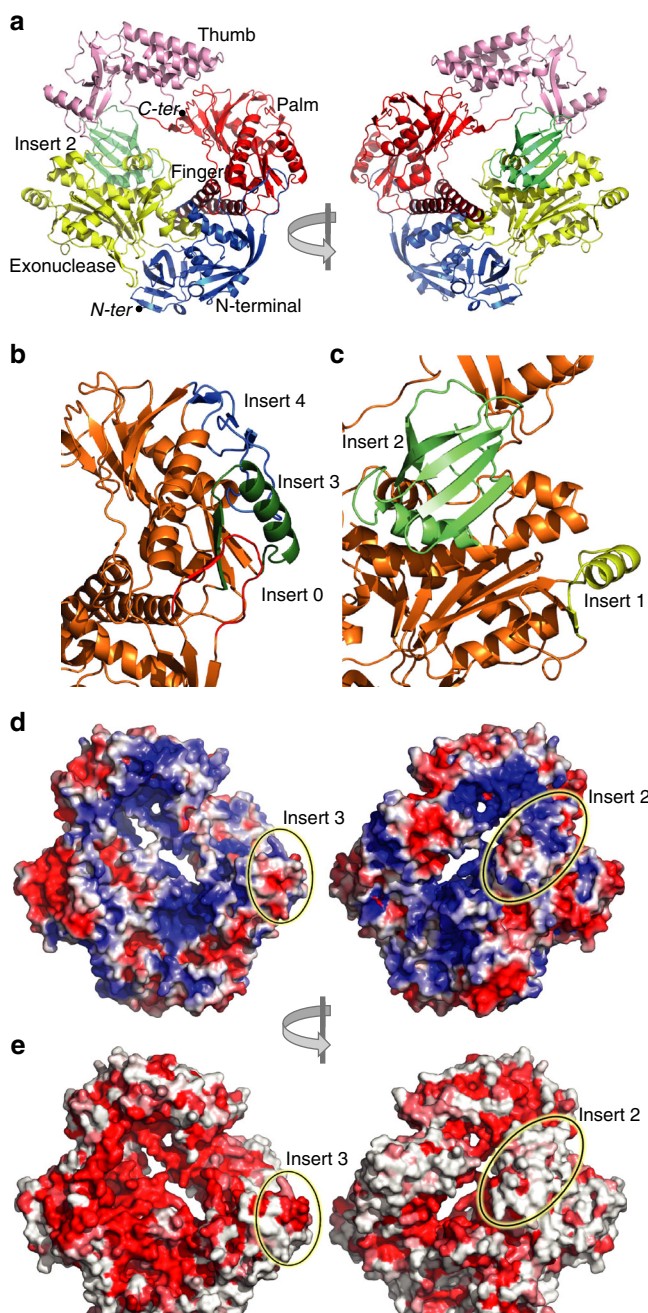

**Fig. 1** Domain structure of E9. Definitions according to Liu et al.[33]: N-terminal: 1–157, 497–523; exonuclease: 158–353, 435–496; insert 2: 354–434; palm: 524–618, 676–829; finger: 619–675; thumb: 830–1006. The same view is used throughout the figure. **a** Front and back view of the domain organization. **b** View of insert 0, insert 3, and insert 4. **c** View of insert 1 and insert 2. **d** View of the electrostatic potential of the solvent accessible surface with colors ranging from red (−3kT/e) to blue (3kT/e). The position of inserts 2 and 3 is indicated. **e** Conservation of residues within 29 representative sequences from the *Chordopoxvirinae* subfamily. Coloring calculated with ESPript[65] as a function of the degree of conservation ranging from red for strict conservation to white. The conserved patch located in insert 3 is encircled as is insert 2, which does not show a particularly conserved surface

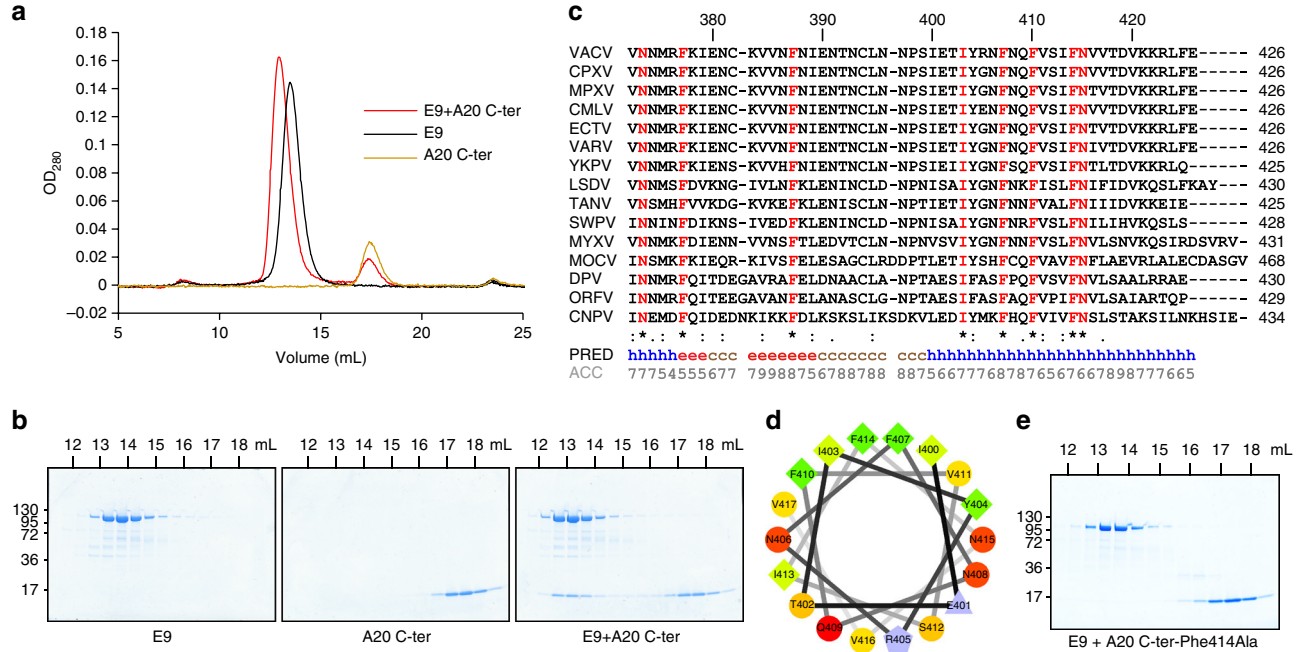

**Fig. 2** The C-terminal domain of A20 (A20 C-ter) interacts with VACV E9. **a** Elution profiles from SEC are compared. A20 C-ter and E9 were either loaded separately or mixed together prior to injection onto the column. **b** SDS-PAGE analysis of the eluted fractions for each run. **c** Alignment of the last amino acids of A20 using representative viruses from each genus of the *Chordopoxvirinae* subfamily: *VACV, CPXV* (cowpox virus), *MPXV* (monkeypox virus), *CMLV* (camelpox virus), *ECTV* (ectromelia virus), *VARV* (variola virus), *YKPV* (yokapox virus), *LSDV* (lumpy skin disease virus), *TANV* (tanapox virus), *SWPV* (swinepox virus), *MYXV* (myxoma virus), *MOCV* (molluscum contagiosum virus), *DPV* (deerpox virus), *ORFV* (ORF virus), *CNPV* (canarypox virus). Conserved residues are in red. The predicted secondary structure using MLRC[66] and its reliability on a 0–9 scale are shown at the bottom of the alignment (ACC). **d** Helical wheel projection with residues of A20 predicted to fold as an α-helix (aa 400–417). Hydrophilic residues are presented as circles, hydrophobic residues as diamonds, charged residues as triangles and pentagons. **e** A point mutation in A20 C-ter abrogates binding to E9. A20 C-ter-Phe414Ala was incubated with WT E9 before injection onto SEC. The eluted fractions were analyzed as in **b**

truncation mutants were isolated by robotic colony picking and tested for expression of soluble protein in an *E. coli* colony-based screen on nitrocellulose filters. Forty-eight clones were selected for scale-up and affinity purification testing. Thirty clones exhibited levels of purified protein visible by Coomassie blue-stained SDS-PAGE. Constructs yielding highly soluble A20 fragments were sequenced and revealed nine unique clones expressing the last 116–148 residues of A20 (Supplementary Fig. 1), of which the clone containing the last 123 residues of the protein (A20 C-ter) was selected for further experiments.

To obtain structural information on this domain, the A20 C-ter fragment was analyzed by SAXS (Supplementary Table 2). The pronounced maximum of the Kratky plot indicated that the construct forms essentially a compact folded structure (Supplementary Fig. 2a). Ab initio modeling suggested a box-shaped molecule with an extension, which might correspond to the additional 24 residues of the biotin acceptor peptide (BAP) and linker used in the ESPRIT screen (Supplementary Fig. 2b, c) and a maximal dimension of 7.4 nm. Additionally, the circular dichroism spectrum of A20 C-ter indicated an α-helix content of about 45% (Supplementary Fig. 2d).

In order to test whether E9 and the A20 C-ter protein fragment interact, purified proteins were analyzed individually and together by size exclusion chromatography (SEC). Loaded individually, E9 and A20 C-ter elute as sharp peaks at 13.5 and 17.4 mL, respectively (Fig. 2a). However, when both proteins are incubated together with a twofold molar excess of A20 C-ter, a first peak is eluted at 12.9 mL and a second one at 17.5 mL (Fig. 2a). SDS-PAGE analysis of the eluted protein fractions showed that the second peak contains the expected excess of free A20 C-ter (Fig. 2b), whereas part of the A20 C-ter is co-eluted with E9 in the first peak indicating an interaction between both proteins.

To confirm the interaction between E9 and A20 C-ter, surface plasmon resonance (SPR) was performed in which A20 C-ter (biotinylated in vivo via its BAP tag) was immobilized on streptavidin-coated chips. Serial dilutions of E9 were injected and analysis of resulting sensorgrams yielded a $K_D$ of 23 nM for the interaction (Supplementary Fig. 3a).

**Identification of residues involved in A20/E9 interaction**. As it was not possible to crystallize the complex between E9 and A20 C-ter, we aimed to characterize the interface using biophysical techniques. The stability of the E9/A20/D4 holoenzyme under high salt conditions (NaCl > 750 mM) observed by us and others[12] suggested that the E9/A20 interaction may be significantly hydrophobic in nature. The alignment of sequence homologs to the A20 C-ter construct encoded by diverse viruses from the *Chordopoxvirinae* subfamily identified only eight strictly conserved residues, all located within the last 54 residues (Fig. 2c). Secondary structure prediction suggested that the last 27 residues form an α-helix with four conserved amino acids (Ile403, Phe407, Phe410, and Phe414) forming a hydrophobic patch in a helical wheel representation (Fig. 2c, d). To determine whether this predicted C-terminal helix of A20 is involved in E9 binding, we constructed three mutants of A20 C-ter in which the three conserved phenylalanine residues were individually changed to alanine (Phe407Ala, Phe410Ala, and Phe414Ala). Bacterial expression of Phe407Ala and Phe410Ala mutants led to insoluble proteins. However, mutant A20 C-ter-Phe414Ala purified like the WT construct and the circular dichroism spectra of WT (Supplementary Fig. 2d) and mutant protein constructs were similar showing 45 and 43% α-helix content, respectively. SEC was performed in order to assess whether the Phe414Ala mutation

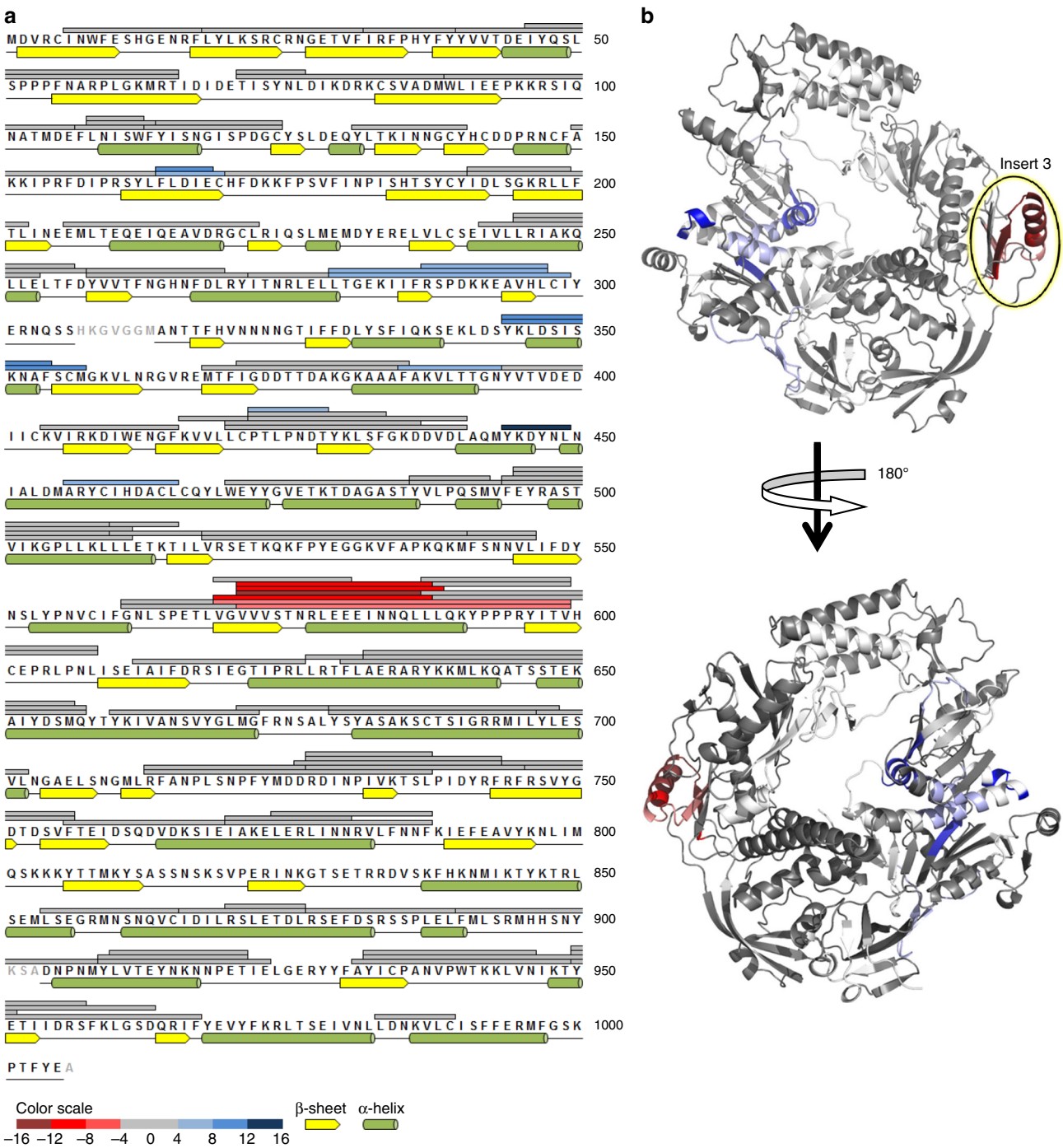

**Fig. 3** Peptides protected by the E9 A20 C-ter interaction determined by H/D exchange MS. **a** Peptides from E9, identified by LC-MS/MS, are represented as bars above the primary sequence. The secondary structure, derived from the crystal structure, is shown underneath the sequence (in gray residues not seen in the structure). The level of protection of individual peptides, as determined by comparing the % D incorporation for free E9 with the one for E9 bound to A20 C-ter, is color coded according to the scale bar. Highly protected areas are shown in red, whereas peptides becoming more exposed upon complex formation are shown in blue. The highly protected region involves the α-helix of insert 3 (residues 577–590). **b** H/D exchange results are mapped onto the crystal structure of E9. Protected residues are color coded as in **a**

affected complex formation between E9 and A20 C-ter. As shown in Fig. 2e, E9 and A20 C-ter-Phe414Ala did not co-purify showing that the interaction was lost. This was confirmed by SPR experiments where the residual interaction between E9 and A20 C-ter-Phe414Ala was too weak to determine a $K_D$ (Supplementary Fig. 3b).

The Phe414Ala mutation was then introduced into the full-length VACV A20. WT and mutant proteins were expressed in insect cells and purified in complex with D4 as previously described[11]. The ability of both heterodimeric complexes to bind to WT E9 was assessed by SPR. WT D4/A20 interacts with E9 with a $K_D$ of 8 nM (Supplementary Fig. 3c) in agreement with previous results[11]. In contrast, D4/A20 Phe414Ala showed much weaker binding and a marked bi-exponential dissociation phase so that a $K_D$ could not be calculated (Supplementary Fig. 3d). Altogether, these data indicated that the putative α-helix present

at the C-terminal extremity of A20 is involved in the interaction with E9.

To identify the interaction surface of A20 C-ter on E9, we used hydrogen–deuterium exchange mass spectrometry under native conditions. We analyzed the E9/A20 C-ter complex by measuring deuterium exchange for 107 partly overlapping peptides from E9, alone, or in complex with A20 C-ter. These correspond to 75% of the E9 primary sequence (Fig. 3a). The comparison of the level of deuteration highlighted one distinct region that showed a strong reduction in deuterium exchange in the complex, indicating protection upon complex formation. Interestingly, all the corresponding peptides cover the α-helix of insert 3 (Figs. 1b and 3b). We also observed peptides displaying increasing deuterium exchange upon A20 C-ter binding, which cluster in the exonuclease domain of E9 and could indicate conformational changes due to the interaction with A20 C-ter (Fig. 3a, b).

In order to confirm the involvement of E9 insert 3 in A20 binding, three mutants of conserved residues in the α-helix were produced. Hydrophobic residues (Leu578 and Ile582) were mutated to alanine (mutant E9-578-582), charged residues (Glu580 and Glu581) were mutated to oppositely charged arginine (E9-580-581) and Gln585-Gln589 and Leu586-Leu588 residues were mutated to Ala and Ser, respectively (E9-585-6-8-9) (Fig. 4a, c, e). When E9-578-582 was incubated with WT A20 C-ter and loaded on SEC, both proteins eluted separately (Fig. 4b), indicating that the interaction is largely reduced; this was subsequently confirmed by SPR ($K_D = 390$ nM, i.e., 17-fold reduction, Supplementary Fig. 3e). When the charged residues were mutated (E9-580-581), the interaction with A20C-ter was still observed on SEC, likewise for the quadruple mutant E9-585-6-8-9 (Fig. 4d, f). However, SPR experiments showed reduced binding: a 10-fold decrease in $K_D$ for the E9-580-581 mutant and fourfold decrease for the E9-585-6-8-9 mutant compared to WT E9 (Supplementary Fig. 3f, g). Thus, we conclude that the hydrophobic residues (Leu578 and Ile582) on the N-terminal side of the insert 3 α-helix are key contacts in E9/A20 complex formation, with neighboring residues involved to a lesser extent, although still enough to permit co-purification of the complex on SEC.

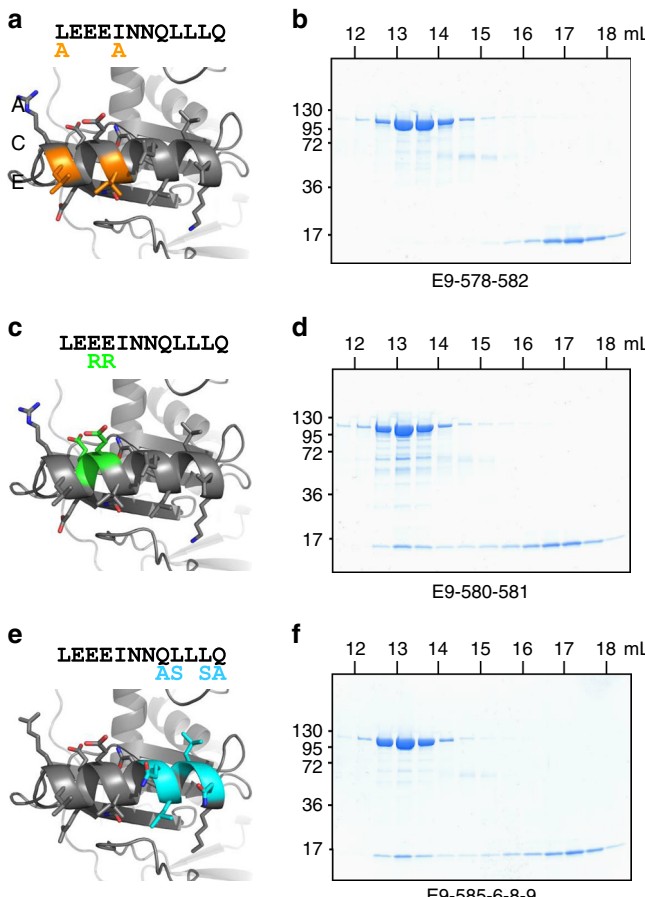

**Fig. 4** Mutational analysis of E9 insert 3 regarding A20 C-ter binding. **a**, **c**, **e** Crystal structure of E9 insert 3 (gray) with mutated residues highlighted. The corresponding sequence of the α-helix of insert 3 (black letters) is shown with mutated amino acids depicted as colored letters. **b**, **d**, **f** Each E9 mutant was mixed with WT A20 C-ter prior to analysis by size exclusion chromatography. A 4–20% SDS-PAGE analysis of the eluted fractions for each of the runs is presented

**Related polymerase structures and modeling of E9/DNA complex.** In order to find the most closely related structures to E9, a PDB search using a block-wise structural alignment was used to overcome the inherent flexibility of polymerases. This yielded family B polymerases including *E. coli* DNA polymerase II (PDB 3k57[30]), archaeal family B polymerases (PDB 2xhb[31]), yeast polymerase δ (pol δ, PDB 3iay[32]), herpes simplex virus (HSV) polymerase (PDB 2gv9[33]), and eukaryotic DNA polymerase α (pol α, for example PDB 4q5v[34]) as the most similar ones. Bacteriophage RB69 polymerase and eukaryotic polymerase ε are more distantly related[35,36]. As the thumb domain movements contribute most to the conformational variability of family B polymerases, a structural alignment of E9 excluding the thumb domain was carried out, which identified yeast pol δ as the most closely related protein (Supplementary Table 1). The close structural relationship between E9 and yeast pol δ was further confirmed when separate E9 domains were superposed individually (Supplementary Table 1).

The published family B polymerase structures in complex with DNA oligonucleotides, with or without an incoming nucleotide mimicking elongation or editing modes, indicate that considerable domain movements occur upon DNA binding leading to a closure of the structure compared to the apo forms. The yeast pol δ structure with bound template and complementary DNA (PDB 3iay) was used to model E9 in elongation mode. We performed SAXS experiments (Supplementary Table 2) using an isomorphous exo$^{minus}$ mutant of E9 (Table 1 and Supplementary Fig. 4a, b) bound to a 29-mer DNA hairpin to check if the best model would be obtained by adjusting only the position of the thumb domain, or if a movement of each individual domain would be required. Complex formation was confirmed by the decrease of the radius of gyration from 3.83 to 3.45 nm upon DNA binding (Supplementary Table 2). The comparison of the SAXS curve of the E9exo$^{minus}$/DNA complex with the calculated scattering of different E9 models (Supplementary Fig. 5a–c) showed that when individual domains of E9 are adjusted (Supplementary Fig. 5c), the theoretical curve fits better (and the $\chi^2$ is lower) than when only the thumb domain is adjusted (Supplementary Fig. 5b) supporting the generalized domain movements upon DNA binding.

Likewise, the enzyme in proofreading or editing mode can be modeled based on the structure of an archaeal polymerase in editing mode[37] (PDB 2xhb, Supplementary Fig. 4c). Larger differences in domain orientation between E9 and the archaeal enzyme make the model globally less reliable, especially since SAXS data are not available. However, the modeled DNA fits very precisely into the exonuclease active site of E9 (Supplementary Fig. 4d), in particular regarding the position of the catalytic Mg$^{2+}$

**Table 2 E9 drug-resistant and temperature-sensitive mutants**

| Mutation | Virus | Effect | Color code | Proposed mechanism | Ref. |
|---|---|---|---|---|---|
| Phe171Ser | VACV | araC[r], araA[r], aph[hs] (when combined with mutant Cys356Tyr or Gly372Asp or Gly380Ser) | Yellow | Modification of 3′-nucleotide binding at exonuclease site | 21 |
| ΔLys174 | VACV | CDV[r] | Yellow | Modification of 3′-nucleotide binding at exonuclease site | 20 |
| His185Tyr | VACV | ts Dts83 | Black | Destabilization of the exonuclease domain | 67 |
| Ala314Thr/ Val | VACV, CMLV[a], MPV[b] | CDV[r] and cross-resistance to other nucleoside phosphonate drugs. Stronger drug-resistance together with mutation Ala684Val. PAA[hs] | Orange | Mutation in beta-hairpin; may modulate guidance from elongation to editing mode | 20,22–26 |
| Ser338Phe | VACV | CDV[r] | Orange | Indirect effect on complementary strand binding or strand switching between elongation and editing mode? | 25 |
| Cys356Tyr | VACV | PAA[r], aph[hs] | Green | Effect on insert 2–finger interaction? | 21 |
| Gly372Asp | VACV | PAA[r], aph[hs] | Green | Effect on insert 2–finger interaction? | 21 |
| Gly380Ser | VACV | PAA[r], aph[hs] | Green | Effect on insert 2–finger interaction? | 21 |
| Gly392Asp | VACV | ts NG26 | Black | Destabilization of insert 2 | 21,68 |
| Ala498Thr/ Val | VACV | aph[r], PAA[hs], araC[hs], araA[hs] | Red | Interference with the rotation of a base in the template required for aph binding | 27 |
| Glu611Lys | VACV | ts Cts42 | Black | Destabilization of insert 3 or of the elongation site | 68 |
| Leu670Met | VACV | aph[r] | Red | Indirect effect on the binding of the template backbone next to aph-binding site | 44 |
| Ala684Val/ Thr | VACV, CMLV[a], MPV[b] | CDV[r] and cross-resistance to other nucleoside phosphonate drugs. Stronger drug-resistance together with mutation Ala314Thr | Red | Indirect effect on the binding of the template backbone in the elongation site | 22–24,26 |
| Ser686Asn | VACV | ts, Dts20 | Black | Destabilization of insert 3 or of the elongation site | 67 |
| Thr688Ala | VACV | CDV[r] and cross-resistance to other nucleoside phosphonate drugs when associated with Ala314Thr mutation. PAA[hs], araC[hs] | Red | Indirect effect on template backbone binding in the elongation site | 22 |
| Thr831Ile | VACV | CDV[r] and cross-resistance to other nucleoside phosphonate drugs | Pink | Modulation of domain movements by a modification at the thumb–palm domain connection and a changed interaction with the complementary strand | 24 |
| Ser851Tyr | VACV | CDV[r] and cross-resistance to other nucleoside phosphonate drugs when combined with Ala684Val mutation | Pink | Modulation of domain movements by a modification of the thumb–palm domain interface | 23 |

r, drug-resistant; hs, drug-hypersensitivity; ts, temperature-sensitive
[a]Camelpox virus
[b]Monkeypox virus

ions replaced by $Mn^{2+}$ in one of the E9 structures (Table 1 and Supplementary Fig. 4e).

## Discussion

The high-resolution structure of the catalytic subunit of the VACV DNA polymerase allows comparison of E9 with other replicative DNA polymerases. Using different criteria such as the global structural similarity of the polymerase domain and structural superpositions of individual subdomains, we found that E9 most closely resembles eukaryotic yeast pol δ. In contrast, RB69 bacteriophage DNA polymerase and eukaryotic polymerase ε are evolutionarily more distant[35,36]. A comparison of the structural elements containing the A20 binding poxvirus-specific insert 3 (i.e., three small β-strands and an α-helix) argues also for a close relationship between E9 and yeast pol δ (Supplementary Fig. 6a, b). The structural features of the equivalent domain in E. coli polymerase II and archaeal polymerases are reduced while the structure is limited to a simple loop in polymerase α. The HSV enzyme is more closely related, underlining parallels of the two representatives of large DNA viruses (Supplementary Fig. 6c). Thus, compared to yeast pol δ and HSV polymerase, the insert 3 of E9 appears to have evolved for cofactor binding, whereas the poxvirus-specific inserts 0 and 4 appear to buttress insert 3 (Fig. 1b).

Insert 2 forms a small 6-stranded β-barrel domain that is also found in ATP synthase F1, EF-Tu, Gar1, and other proteins[38,39]. It only shows weak sequence conservation between poxviruses (Fig. 1e) and the absence of strong electrostatic features (Fig. 1d) make it unlikely that this domain interacts with nucleotides. Interestingly, insert 2 carries a number of resistance mutations toward PAA (Table 2) that could reveal a possible contact between this domain and the finger domain in the presence of an incoming nucleotide (discussed below).

Analysis of the E9 structure does not explain the unique role of the enzyme in recombination[17], as no obvious domain could be related to such an activity. Mechanistically, the reaction was shown to require the 3′-to-5′ exonuclease activity of the polymerase[18]. It is intriguing that insert 2 is located within the essential exonuclease domain. However, poxvirus DNA synthesis and recombination appear to be tightly linked processes and, to date, genetic and biochemical studies have failed to isolate E9 mutants with functionally separate activities.

A low-resolution structure of the holoenzyme obtained by SAXS is available[11]. The E9 structure can be positioned unambiguously in the large part of this envelope but the orientation of the disk-shaped E9 around its short axis remains unclear (Fig. 5a). Using manual fitting based on the position of the A20-binding site on E9 (i.e., the α-helix from insert 3), the orientation of E9 can be obtained with an accuracy of about ±40°. Using the

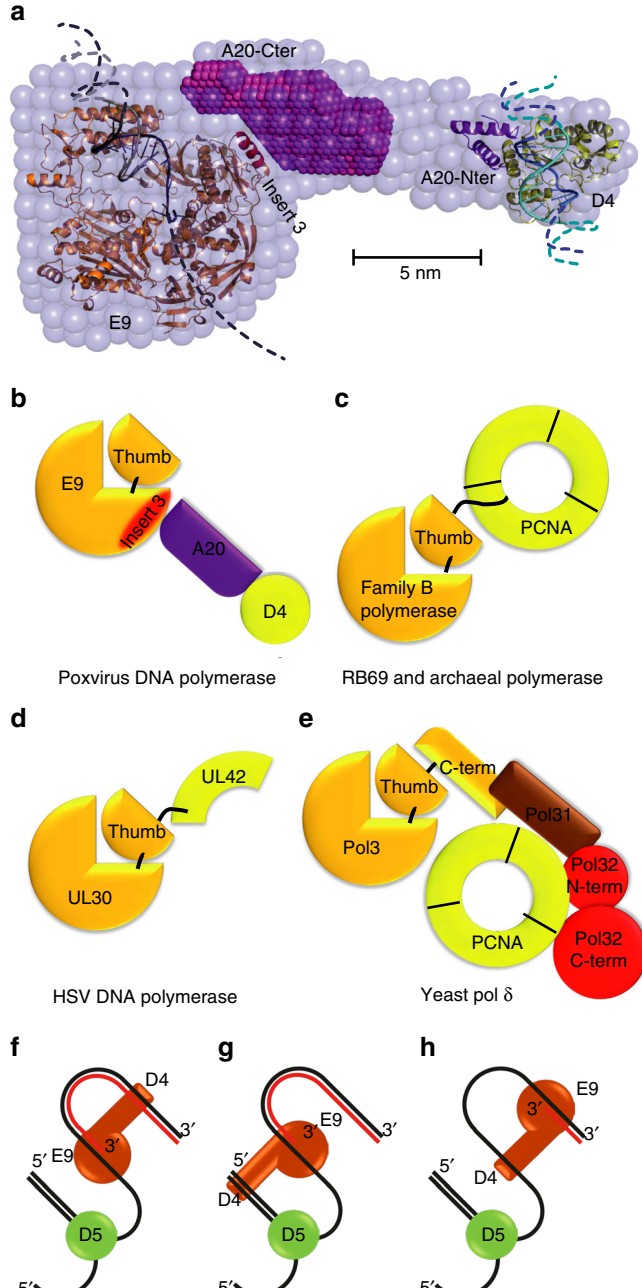

**Fig. 5** Models of polymerase holoenzymes. **a** Model of the VACV holoenzyme. The envelope obtained by SAXS[11] is used to define the global outline. Components of the polymerase holoenzyme have been placed manually: a model of E9 in complex with DNA in elongation mode, the D4/A20$_{1-50}$ complex with bound DNA (A20$_{1-50}$, in violet, D4 in yellow, and DNA in blue)[13], the SAXS ab initio model of the C-terminal fragment of A20 (magenta). Schematic view of the processivity factor binding of **b** E9, **c** archaeal and phage DNA polymerases, **d** HSV DNA polymerase, and **e** S. cerevisiae polymerase δ. Different orientations of the holoenzyme (brown) at the replication fork, where D4 binds either the newly synthesized dsDNA strand (**f**), or the incoming template strand before (**g**) or after (**h**) strand separation by the helicase–primase D5 (green)

same approach, the model of D4/A20$_{1-50}$ in complex with dsDNA[13] can be positioned at the other extremity with the DNA oligonucleotide being almost perpendicular to the long axis of the holoenzyme. The orientation of D4 around this axis cannot be defined.

In general, family B polymerases require co-factors for processivity. Figure 5b-e illustrates the evolution of various co-factors bound to their cognate polymerase. For the bacteriophage RB69 and archaeal polymerases, a direct interaction between the trimeric sliding clamp processivity factor and the extreme C terminus of the DNA polymerase is observed[40] (Fig. 5c). Similarly, HSV polymerase interacts with its PCNA-related processivity factor (UL42) through the C-terminal peptide of the polymerase[41] (Fig. 5d). Eukaryotic polymerase δ (pol3 in yeast) carries an elongated C-terminal domain that binds to PCNA[42] and to an additional subunit, pol31. Pol31 interacts with pol32 whose C-terminal domain binds also to PCNA (Fig. 5e). A low-resolution structure shows that the three-dimensional arrangement of the yeast pol3, pol31, and the N-terminal domain of pol32[43] resembles the E9 holoenzyme[11].

As the C terminus, with the exception of the last three residues, is involved in the stabilization of the E9 thumb domain structure, it is unlikely to interact through a linear peptide motif with other proteins, e.g., A20 or PCNA. By consequence, E9 neither binds to its co-factor through a peptide at the C terminus of the relatively mobile thumb domain (as observed in herpesviruses, RB69 and archaeal polymerases) nor does it possess a C-terminal domain interacting with additional subunits (as eukaryotic δ polymerases). Instead, the principal-binding site for A20 (around the poxvirus-specific insert 3) is located relatively close to the C terminus of E9, but in the palm domain (Fig. 5a, b), although additional interactions with other domains cannot be fully excluded. To date, it is not possible to infer any relationship between VACV A20 and polymerase δ co-factor subunits or any other protein at the level of sequence or secondary structure. Understanding the evolution of A20 as a necessary subunit of the E9 processivity factor will require the high-resolution structure of the full-length protein.

There are two possibilities for the orientation of the VACV holoenzyme at the replication fork: either D4 interacts with the newly synthesized dsDNA (Fig. 5f) or D4 binds to the parental ss/dsDNA (Fig. 5g, h). In the latter case, as there is no information on the actual length of the DNA between the DNA-binding site of D4 and the active site of E9, it is possible that the D5 helicase–primase is located between both binding sites (Fig. 5g). It cannot be excluded that D5 moves ahead of the polymerase holoenzyme with D4 binding to ssDNA (Fig. 5h) as D4 appears to bind ssDNA and dsDNA equally[13]. Further biochemical and structural work will allow a better understanding of the holoenzyme/DNA complex organization at the replication fork.

Several antiviral molecules targeting E9 have been described and various resistance mutations against these drugs have been selected and characterized (Table 2). These inhibitors (PAA, aph, AraC, the large DNA virus-specific molecule CDV, and some related compounds[23]) display broad-spectrum activities against family B polymerases. PAA (a pyrophosphate analog) and aph (competing with dCTP) are inhibitors blocking polymerase action, whereas AraC and cidofovir (both nucleoside analogs) terminate chain elongation after their integration into newly synthesized DNA. The modeled E9 structure in a closed, DNA-bound conformation suggests three main resistance mechanisms for CDV, AraC, and aph (Fig. 6a): (i) changes in the 3′-nucleotide binding in the exonuclease site facilitating the removal of chain terminators, (ii) perturbation of template backbone binding in the elongation site by modification of residues in the hydrophobic core impeding incorporation of the nucleotide analog or binding of aph, and (iii) mutations affecting the switch from elongation to editing mode.

(i) The AraC resistance mutation Phe171Ser[21], located in the exonuclease active site, targets a residue predicted to contact the 3′-nucleotide of the DNA (Fig. 6a and Supplementary Fig. 4d).

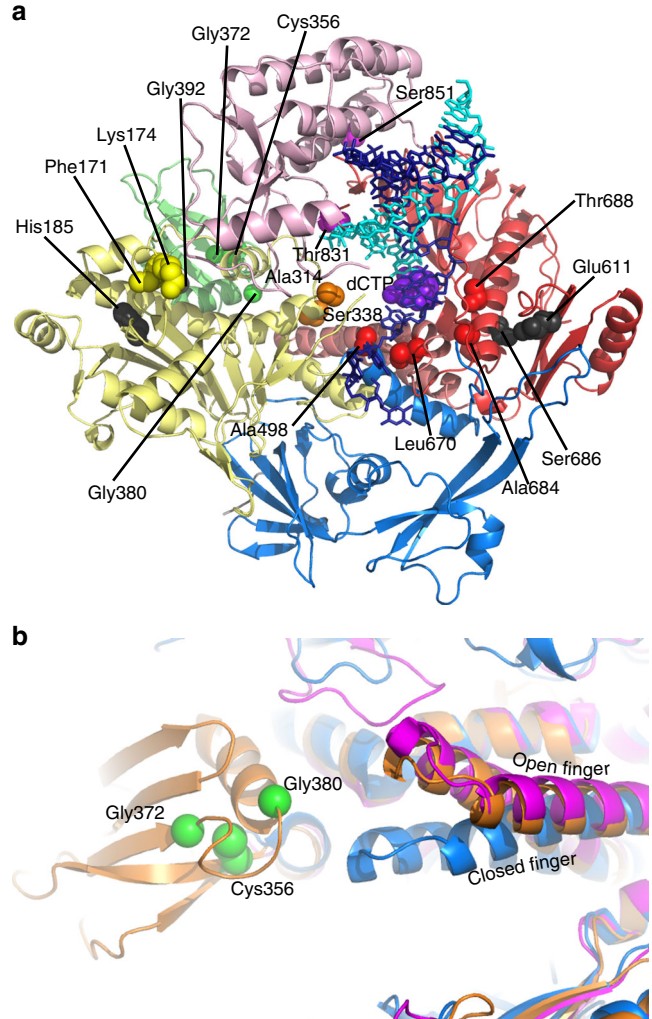

**Fig. 6** Analysis of drug-resistance mutations on E9. **a** Locations of different mutants listed in Table 2 on the model of E9 in complex with DNA in elongation mode. Domains of the polymerase are color coded as in Fig. 1a. **b** Analysis of the finger domain movements: in magenta, the DNA-bound structure of human pol α (PDB 5iud), in orange E9 after superposition of the palm and exonuclease domains onto the binary complex of human pol α, in blue the ternary complex of yeast pol δ with DNA and an incoming nucleotide (PDB 3iay). Upon nucleotide binding, the tip of the finger of E9 may move close to the insert 2 domain allowing a potential contact, which would occur in vicinity of the PAA resistance mutations depicted as green spheres

Similarly, the cidofovir-resistant deletion mutant ΔLys174 will destabilize Phe175, which may also contact the 3′-base. The changes in the recognition of the 3′-nucleotide may affect indirectly the hydrolysis of the modified DNA backbone by influencing its position.

(ii) The aph resistance mutation Ala498Thr/Val[27] is likely to block the template guanine base in a position interfering with aph binding. In close proximity, the Leu670Met aph resistance mutation[44], the CDV resistance mutations Ala684Val/Thr[22–24,26], and the CDV resistance mutation Thr688Ala[22] may modify the packing of hydrophobic residues next to the template strand-binding site. This may lead to a distortion of the template strand conformation disfavoring the integration of CDV diphosphate[23], or may facilitate further elongation following CDV incorporation at the N-2 position[45].

(iii) The prominent β-finger located in the exonuclease domain (residues 299–319) is a structural element, which seems important for switching between elongation and editing modes as it is able to contact the template strand during elongation and intervene in strand separation required for proofreading[46]. In E9, this domain harbors the principal CDV resistance mutation Ala314Thr/Val[20,22–26] that facilitates excision of nucleoside analogs[18]. Indirectly, the Ser338Phe mutation[25] may also affect the switch from elongation to editing mode. Likewise, Thr831Ile and Ser851Tyr mutants[23,24] located at the interfaces of the thumb domain may influence the domain movements required for transition from elongation to editing mode facilitating indirectly the excision of nucleotide analogs. The Thr831Ile mutation might also interact directly with the complementary strand.

The resistance mutations raised against PAA[21] are a class on their own (Fig. 6b). They are all located in the poxvirus-specific insert 2 domain and could only be explained by a movement of the finger domain upon binding of an incoming nucleotide as described for other family B polymerases[47,48]. This would bring the tip of the finger and insert 2 into contact (Fig. 6b). The same movement has been observed upon binding of the PAA-related molecule phosphonoformate to a modified RB69 polymerase mimicking the finger domain from HSV, which blocks the polymerase in a closed conformation[49]. Consequently, mutations Cys356Tyr and Gly380Ser that likely destabilize the hydrophobic core of the insert 2 domain, or Gly372Asp that may affect the position of the surface residue Trp411, could have an indirect effect on the interaction of insert 2 with the finger, which in turn might influence PAA binding in the active site.

Altogether, the high-resolution structure of the poxvirus DNA polymerase not only identifies the mode of processivity factor binding, but also permits an understanding of antiviral resistance mechanisms, although the dynamic character of the polymerase still requires further structural work in order to obtain more precise snapshots of the different functional states. The identification of the E9/A20 interface will facilitate the design of compounds or peptides that disrupt this interaction, whereas the high-resolution structure of the polymerase active site will accelerate the development of new antiviral drugs.

## Methods

**E9 expression and purification**. VACV E9 (Copenhagen strain) was expressed in high Five™ insect cells (Thermo Fisher Scientific) infected with a recombinant baculovirus carrying the *E9L* gene fused to a N-terminal His-tag and a TEV cleavage site[11]. Cell suspensions were grown in Express Five-SFM medium (Gibco) at 27 °C following protocols described in Trowitzsch et al.[50]. Cell pellets were resuspended in 10 volumes of equilibration buffer (50 mM Tris-HCl pH 7, 300 mM NaCl, 10 mM β-mercaptoethanol, and 10 mM imidazole) with cOmplete, EDTA-free protease inhibitor cocktail (Roche). Cells were disrupted mechanically using a Potter-Elvehjem homogenizer. The lysate was clarified by centrifugation at 48,000×*g* for 20 min at 4 °C. The supernatant was loaded onto a 5 mL HisTrap FF crude column (GE Healthcare) previously equilibrated with equilibration buffer. The column was washed with equilibration buffer containing 20 mM imidazole and proteins were eluted with 100 mL of a linear 20–250 mM imidazole gradient. Fractions containing E9 were pooled and concentrated using Amicon centrifugal filter units (Millipore). The buffer was exchanged to equilibration buffer using an Econo-10 DG desalting column (Bio-Rad). The protein was digested overnight at 20 °C with tobacco etch virus (TEV) protease at a ratio of 1/100 (w/w) and was loaded onto a 5 mL HisTrap FF crude column. E9 was recovered in the flow-through fraction and was concentrated prior to injection onto a Superdex 200 GL 10/300 column (GE Healthcare) equilibrated with gel filtration buffer (20 mM Tris-HCl pH 7, 300 mM NaCl, 4 mM dithiothreitol (DTT)). Fractions were analyzed by SDS-PAGE and stained with InstantBlue (Expedeon). The E9 mutants subsequently called exo^minus (Asp166Ala + Glu168Ala) (exonuclease deficient by mutation of Mg²⁺ coordinating residues), E9-578-582 (Leu578Ala + Ile582Ala: aliphatic residues replaced by small residues), E9-580-581 (Glu580Arg + Glu581Arg: charge reversal), and E9-585-6-8-9 (Gln585Ala + Leu586Ser + Leu588Ser + Gln589Ala: aliphatic residues replaced by small polar residues, large polar residues replaced by small residues) were produced by PCR-based site-directed mutagenesis (Supplementary Table 3) and expressed and purified as described above.

**E9 crystallization and data collection**. The E9 protein was concentrated to 7 mg mL$^{-1}$ in 20 mM Tris-HCl pH 7, 300 mM NaCl, 4 mM DTT using Amicon 50 kDa concentrators. Initial crystallization conditions giving diffracting crystals were found in the Morpheus screen (Molecular Dimensions) (10% PEG 4000, 20% glycerol, 100 mM MES-imidazole pH 6.5, 15 mM MgCl$_2$, 15 mM CaCl$_2$) using the EMBL Grenoble high-throughput crystallization facility and were refined manually to 9–11% PEG 3000, 20–25% glycerol, 100 mM MES-NaOH pH 6.25. Heavy atom derivatives were obtained by soaking crystals for 2–24 h in reservoir solution complemented with 1 mM of either Pb(CH$_3$COO)$_2$ or GdCl$_3$. Needle-like crystals were flash frozen in liquid nitrogen before data collection on ESRF beamlines ID23-1 for native data sets and BM14 for heavy atom derivatives. Helical data collections were used to increase the exposed sample volume in order to overcome radiation damage. Mn$^{2+}$ derivatised crystals were obtained by soaking native crystals in the reservoir solution with an additional 5 mM MnCl$_2$.

**Structure determination**. The structure of E9-WT was determined using the MIRAS method using optimized anomalous scattering. XDS[51] was used for data integration, AIMLESS[52] was used for data reduction, SOLVE and RESOLVE[53] were used for phase determination and improvement. A first model built with BUCCANEER[54] was further refined using cycles of manual inspection and building using COOT[55] and restrained refinement with individual B factors but without translation/libration/screw refinement using REFMAC5[56]. E9exo$^{minus}$ and Mn$^{2+}$ structures were isomorphous.

**Structure analysis and modeling**. The final E9 structure was compared to other DNA polymerases using the flexible structure alignment algorithm implemented in FATCAT[57]. The fold of the insert 2 domain was identified with CATH[58]. Modeling of the closed and DNA-bound conformations used the "super" structural alignment function of PyMOL (The PyMOL Molecular Graphics System, Version 1.4.1 Schrödinger, LLC). Figures were generated with PyMOL.

**Identification of the A20 C-ter soluble construct**. The full-length 1281 bp VACV *A20R* gene was subcloned into pESPRIT002, a pET9a-derivative encoding N-terminal His-tag and C-terminal BAP. For 5′-exonuclease III/mung bean nuclease truncation libraries, AatII and AscI sites are positioned at the 5′ end of the insert; the 3′ end was cloned in-frame with the BAP-encoding sequence. The plasmid was processed through truncation reactions and solubility screening as described[29,59]. Briefly, pESPRIT002-A20 was digested with both AatII and AscI. Four µg of linearized plasmid (in 60 µL reactional volume) was digested with exonuclease III and 1 µL aliquots quenched into a single tube each 60 s to generate nested deletions. The single-stranded overhangs were removed with mung bean nuclease and the ends of the plasmid polished with *Pfu* polymerase. The truncated plasmid mix was electrophoresed on an agarose gel and two bands corresponding to plasmid with A20 inserts of ~0–600 and 600–1281 bp were excised. Plasmids were purified from gel slices and recircularized with T4 DNA ligase. Competent *E. coli* MACH1 cells (Thermo Fisher Scientific) were transformed with the two sublibraries and the insert sizes determined by colony PCR with standard T7for and T7rev primers. Approximately 10,000 colonies for each sublibrary were harvested from transformation plates and plasmid purified from cell pellets. Electrocompetent *E. coli* BL21 AI cells (Thermo Fisher Scientific) were transformed with the 0–600 and 600–1281 bp sublibraries and plated on LB agar trays. Colonies from small and large insert sublibraries were picked robotically into HMFM-TB broth with antibiotics in 12 × 384-well plates (~4600 individual clones). These were grown overnight at 37 °C, replicated into fresh TB, grown to saturation, then arrayed robotically at high density onto nitrocellulose membranes on LB agar with antibiotics. Plates were incubated at 37 °C until colonies were just visible, then nitrocellulose membranes bearing arrayed colonies were transferred to fresh LB agar plates with antibiotics, biotin, and arabinose at 30 °C for 4 h for protein expression. Putative soluble constructs were identified by in situ lysis of colonies and colony blotting using streptavidin Alexa488 (Thermo Fisher Scientific) and anti-His-tag mouse monoclonal antibody (GE Healthcare) followed by Alexa532-labeled rabbit anti-mouse secondary antibody (Thermo Fisher Scientific). Membranes were scanned using a Typhoon Trio imager (GE Healthcare) with analysis of images using VisualGrid software (GPC Biotech) with data analysis in a spreadsheet. The 48 highest ranking clones were isolated from the library, grown and induced in 4 mL TB cultures, then His-tagged proteins were purified on Ni-NTA agarose (Qiagen) and analyzed by SDS-PAGE and western blot against the His-tag. Construct boundaries of expression clones were characterized by DNA sequencing with standard T7for and T7rev primers.

**A20 C-ter expression and purification**. In order to express A20 C-ter, *E. coli* Rosetta (DE3) pLysS strain (Novagen) was transformed with the pESPRIT002-A20 C-ter vector. Cultures were grown at 37 °C in LB Broth medium (Sigma) in the presence of kanamycin (25 µg mL$^{-1}$) and chloramphenicol (34 µg mL$^{-1}$) until the OD$_{600}$ reached 0.4–0.6. Protein expression was then induced with 0.1 mM of isopropyl β-D-1-thiogalactopyranoside. Bacterial growth was pursued at 18 °C for 16 h. Bacteria were centrifuged and resuspended in binding buffer (25 mM Tris-HCl pH 7.5, 300 mM NaCl, 20 mM imidazole, 5 mM β-mercaptoethanol) and cOmplete, EDTA-free protease inhibitor cocktail (Roche). Cells were lyzed by three

freeze-thaw cycles and sonication. Cell lysate obtained after centrifugation at 48,000×*g* for 20 min was loaded onto a 1 mL HisTrap FF crude column (GE Healthcare) equilibrated with binding buffer. The column was washed in the same buffer containing 50 mM imidazole and the protein was eluted with 200 mM imidazole. Fractions containing A20 C-ter were pooled and desalted on a PD10 column (GE Healthcare) in 25 mM Tris-HCl pH 7.5, 300 mM NaCl and the His-tag was cleaved by TEV protease. A20 C-ter was then recovered from the flow-through of a 1 mL HisTrap FF crude column. Proteins were further loaded on SEC (Superdex 75 10/300 GL, GE Healthcare) equilibrated in 25 mM Tris-HCl pH 7.5, 300 mM NaCl. Fractions were analyzed by SDS-PAGE and stained with InstantBlue (Expedeon). The A20 C-ter-414 mutant (produced by PCR-based site-directed mutagenesis, Supplementary Table 3) was expressed and purified using the same protocol. Circular dichroism spectra were recorded with 0.2 mg mL$^{-1}$ of A20 C-ter in 300 mM NaF, 50 mM potassium phosphate pH 7.5 using a JASCO J-180 spectropolarimeter and analyzed on the Dichroweb website using the Contin[60] algorithm.

**Hydrogen/deuterium exchange mass spectrometry**. H/D exchange experiments were performed using a fully automated system consisting of PAL autosampler (CTC Analytics) and a custom-built Peltier-cooled box, which contained two Rheodyne valves mounted with an immobilized pepsin column (2 × 20 mm), a trap cartridge (Trap Acquity UPLC Protein BEH C18 2.1 × 5 mm, Waters) and an analytical column (Acquity UPLC BEH C18 1.7 µm 1 × 100 mm, Waters). The whole system was maintained at 4 °C.

H/D exchange was followed for E9 or A20 C-ter alone or their equimolar mixture. Deuteration was started by mixing 10 µL of a protein sample with 40 µL of deuterated buffer. After 2 min, the exchange was quenched by the addition of 50 µL of 200 mM glycine-HCl, pH 2.3 and the proteins were immediately injected onto the LC system described above. Online digestion on immobilized pepsin column with subsequent peptide desalting on a trap column lasted 3 min and was driven by a flow of 0.4% formic acid in water (solvent A, flow rate of 100 µL min$^{-1}$). Next, peptides were separated with a gradient elution from 15 to 70% solvent B (0.4% formic acid in 95% acetonitrile) during 10 min followed by 1 min at 100% solvent B. The gradient was produced by an UPLC pump (Agilent Technologies) operating at 50 µL min$^{-1}$. The outlet of the analytical column was directly interfaced to an ESI source of TOF mass spectrometer (Agilent 6210). Spectra were collected in a positive ion mode over the mass range 300–1300 *m/z*. Data were interpreted using HD Examiner software (Sierra Analytics). All experiments were conducted in triplicate.

Peptides generated upon online digestion were analyzed by LC-MS/MS prior to the H/D experiment. For the gradient, pepsin column and flow rates were identical to the setup described above. Exceptions in the LC-MS setup were as follows: a desalting cartridge peptide Opti-Trap Micro from Optimize Technologies and an analytical column Jupiter C18, 0.5 × 50 mm, 5 µm, 300 Å from Phenomenex was used. The gradient on the analytical column lasted 30 min with a flow rate of 15 µL min$^{-1}$. The outlet of the column was connected to the ESI source of a 15T FT-ICR instrument (solariX XR) operating in data-dependent mode, where each MS scan was followed by six MS/MS scans (CID in a quadrupole). Data were searched by the MASCOT algorithm against a database containing protein sequences of E9, A20 C-ter, and porcine pepsin A.

**D4/A20 expression and purification**. VACV D4 (fused to a N-ter His-tag and a TEV cleavage site) and full-length WT or mutant A20 were co-expressed in insect cells infected with a recombinant baculovirus as described in Sèle et al.[11]. Protein expression and purification were essentially performed following the protocols described for E9, except the elution step from the first nickel column, which used 100 mM imidazole. The final gel filtration step was performed on a Superdex 200 GL 10/300 column (GE Healthcare) equilibrated in 25 mM Tris-HCl pH 7.5, 300 mM NaCl. The D4/A20 Phe414Ala mutant was generated by PCR-based site-directed mutagenesis (Supplementary Table 3).

**Surface plasmon resonance**. SPR acquisitions were carried out on CM5 sensorchips on a BIAcore 3000 instrument (GE Healthcare). All experiments were performed in buffer containing 100 mM NaCl, 25 mM Tris-HCl pH 7.5 at a flow rate of 15 µL min$^{-1}$. For experiments involving A20 C-ter, about 2500 resonance units (RU) of streptavidin (Sigma-Aldrich) were immobilized on the EDC-NHS activated surfaces. A20 C-ter (or A20 C-ter-Phe414Ala) was injected at 10 µg mL$^{-1}$ into one flow cell until ~1000 RU were reached. A second flow cell (without bound A20 C-ter) was used for background subtraction. Twofold serial dilutions (160–5 nM) of E9 in running buffer were injected during 180 s (association phase) followed by a 150 s dissociation phase. Similarly, for experiments involving full-length D4/A20 complex or Phe414Ala mutant complex, E9-WT was immobilized and a twofold serial dilution (160–5 nM) of complex in running buffer was injected. Background subtracted signals were exported from the Biologic software (GE Healthcare) and imported into LibreOffice Calc (www.libreoffice.org) for curve fitting using the Solver function and figure preparation.

**Small-angle X-ray scattering experiments**. An E9exo$^{minus}$/DNA complex has been prepared by mixing E9 exo$^{minus}$ with a 20% molar excess of a DNA 29-mer

forming an hairpin structure with five bases overhang at the 5′ end, 10 base-pairs and a 4 nucleotide loop (5′-AAAGGCGCTGCTGAGTTTTCTCAGCAGCG-3′) similar to the one used by Killilea and co-workers[37]. Fifty μL of purified E9 (4 mg mL⁻¹), or E9exo$^{minus}$ with 29-mer DNA oligonucleotide were injected onto a Superdex 200 Increase 5/150 GL column (GE Healthcare) in-line with the flow cell for SAXS[61] equilibrated with 20 mM Tris-HCl, pH 7.5, 100 mM NaCl. For A20 C-ter, 30 μL at 5 mg mL⁻¹ were injected onto a Superdex 75 3.2/300 (GE Healthcare) in 25 mM Tris-HCl, pH 7.5, 300 mM NaCl. Runs were performed at a flow rate of 0.3 mL min⁻¹ and 3000 frames of 1 s were collected using a Pilatus 1M detector (Dectris). Individual frames were processed automatically and independently within the EDNA framework[62] yielding radially averaged curves of normalized intensity vs. scattering angle $s = 4\pi\sin\theta/\lambda$. Frames corresponding to the elution of the protein of interest were identified in iSPyB[63], merged and analyzed further using the tools of the ATSAS package[64]. For A20 C-ter 40 ab initio models were calculated using DAMMIF, averaged, aligned, and compared using DAMAVER. The agreement between scattering curves of E9 and E9exo$^{minus}$/DNA complex and atomic models were calculated using CRYSOL. Curves were plotted with MS Excel.

**Data availability**. Coordinates and structure factor amplitudes have been deposited in the PDB as entries 5N2E for E9, 5N2G for the Mn$^{2+}$ complex, and 5N2H for the E9exo$^{minus}$ mutant. SAXS data and models have been deposited in the SAXSBDB as entries SASDCM5 for the E9exo$^{minus}$/DNA complex, SASDCN5 for E9, and SASDCP5 for A20 C-ter. Other data supporting the findings of this study are available from the corresponding author on reasonable request.

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

## Acknowledgements

We acknowledge financing of the project by the French Grants REPLIPOX ANR-13-BSV8–0014 and by research grants from the Service de Santé des Armées and the Délégation Générale pour l'Armement. P.M. acknowledges support from NPU II (LQ1604). This work used the platforms of the Grenoble Instruct-ERIC Center (ISBG: UMS 3518 CNRS-CEA-UGA-EMBL) with support from FRISBI (ANR-10-INSB-05–02) and GRAL (ANR-10-LABX-49-01) within the Grenoble Partnership for Structural Biology (PSB). Special thanks to F.C.A. Gerard for her help with MALLS-RI and P. Fender for the help with Biacore experiments. We are grateful toward the ESRF for beamtime and thank in particular H. Belrhali, A. McCarthy, and M. Brennich for support. We further wish to acknowledge F. Garzoni from the Berger Group at EMBL Grenoble for his kind help with the insect cell culture.

## Author contributions

N.T. and F.I. designed research; N.T. and F.I. performed the main experiments with the help of C.D.; P.J.M. and D.J.H. performed ESPRIT on A20. P.M. and E.F. carried out the MS experiments. S.H. performed SAXS measurements and analysis. N.T., C.N.P., W.P.B. and F.I. analyzed data; N.T., W.P.B. and F.I. wrote the paper with input from all the authors.

## Additional information

**Competing interests:** The authors declare no competing financial interests.

