## [Peer Review file · Nature Communications]

Reviewers' comments:

Reviewer #1 (Remarks to the Author):

The manuscript by Tarbouriech et al. presents the first crystal structure of the vaccinia virus DNA polymerase, and as such is a major advance in the field. For all viruses, the polymerase is the core of the replication machinery, a major determinant of genetic stability and variability, and a significant therapeutic target. As could have been predicted from sequence comparisons, the polymerase is a member of the B family of polymerases and its structure has the classic palm, thumb and finger components. Superimposed on this similarity, however, is the presence of several poxvirus-specific inserts that this same group noted in earlier publications. The current work provides new insight into the structure of these inserted sequences.

Another unique feature of the vaccinia polymerase is the nature of its processivity factor, a heterodimer of the A20 and D4 proteins. A20 binds to both the polymerase and D4, and serves as a bridge between these two proteins. Until now, there has been no insight into which region(s) of Pol bind to A20 and which region(s) of A20 bind to Pol. In this work, the authors determine that the C'-terminal 123 residues of A20 are sufficient to bind to Pol with a K_d of 23 nM. They also determine that the region of Pol that binds to A20 is the poxvirus-specific "insert 3" region. This finding is significant in that for many/most related polymerases, it is their C' terminus that binds to their processivity factor. For both A20 and Pol, residues that contribute to the interaction were identified using preliminary site-specific mutagenesis. Thus, this structural analysis of Pol has already enabled significant advances in the understanding of how the holoenzyme is assembled.

The work is well done, utilizes a number of different and effective techniques, and allows important conclusions to be drawn. The work is important and will be of significant interest to both poxvirologists and those interested in polymerase structure/function. A few comments should be addressed:

- Czarnecki and Traktman (2017, *Virus Research*, PMID: 28159613) published a thorough review of the E9 DNA polymerase and the A20/D4 processivity factor earlier this year (before the current manuscript was submitted), with significant detail about genetic lesions and structural analysis. The authors should compare their mutant assignments and phenotype assignments with those in this recent review. The omission of this reference is a serious flaw that must be corrected.
- The authors cite the work from the Evans laboratory showing that the vaccinia Pol is unusual in being able to mediate strand annealing reactions (in association with the single-strand binding protein I3). Does their structure provide any insight into this property ?
- For a broader non-virological audience, it would be useful to have some general references for summary statements made in the Introduction about the sites(s) of viral DNA replication, etc.
- The list of proteins that are essential for viral replication is somewhat incomplete. For example, the viral DNA ligase (A50) appears to be vital for replication in the absence of the cellular ligase I.
- On page 12, the abbreviation "HCV" is incorrect and should be changed to "HSV", for herpes simplex virus (not hepatitis C virus).
- On page 6, while describing the preparation of a library of A20 fragments, the manuscript has the phrase "Approximately 4 600 clones were isolated and tested". The meaning of this phrase is unclear and should be corrected.

Reviewer #2 (Remarks to the Author):

The manuscript entitled 'Structural basis of the novel mode of processivity factor binding by vaccinia virus DNA polymerase' by Tarbouriech et al describes the X-ray structure of the catalytic subunit of the DNA polymerase E9 and locates on the palm domain a binding site for the C-terminal residues of the A20 processivity factor. From here the authors then initiate a comparative analysis with other DNA polymerase species and infer mechanisms of drug resistance.

General Comments

The article reads amenably. It's nicely tailored as it combines complementary structural and biophysical techniques and it addresses an important area of research. However the breadth of the findings on which the discussion and conclusion are based, is questionable. The crystallographic study of the E9 DNA polymerase is solid although some minor revisions are required in the Table 1 (see below).

The section on the identification of a soluble A20 (49 kDa) domain interacting with E9 poses some questions. While the experimental part (ESPRIT technology) leading to this identification has proved a successful strategy to recognize soluble portions of a protein, the major concern is that the found A20 C-ter might not be the only region interacting with the E9 polymerase. The evidence on which the interaction between E9 and the C-terminal domain of A20 is hypothesized comes from SAXS studies on the D4-A20-E9 complex that provided an envelope in which the known/available structures at the time were manually fitted (see Fig. 8 in Ref 9) – there are no other biochemical or cross-link mass-spec expts on the E9 and full-length A20 complex, for example, that would further strengthen/corroborate this and map possible interacting residues between the two subunits, including those assumed with the A20 C-ter. And while it has been proved that the A20 C-ter and E9 interacts via gel chromatography and SPR (and further in the text identified the putative interacting residues via LC-MS/MS) – data supporting the exclusiveness of this C-ter of the A20 region are limited, a fact that undermines the generality of the findings of this aspect of the work.

Moreover, using SAXS *ab initio* on a 123 residues protein fragment (A20 C-ter) might be stretching the limit of envelope interpretation (as also the fitting of a SAXS envelope in another previously determined SAXS enveloped, Fig 5a). Regarding this (A20 C-ter) it would have been useful to explain in the SI what restraints were used that led to the final 'best' model envelope (Suppl. Fig 2), and of course to show the second and third best solution.

As for the identification of the interacting residues the described logic is solid and linear with the combination of mutagenesis and mass-spec techniques and yet the entire framework is built on a secondary structure prediction of the last 27 residues of the A20 C-ter being α -helical (BTW there is no confidence scale or reliability score in panel c of Fig. 2). As this is an important assumption, much stronger and further experimental data would have been beneficial to prove the case. For example, why not synthesise the last 30 residues and check, for example, by CD the helical propensity of the simple fragment, or as the A20- C-ter is produced in bacteria why not collaborate with a NMR group and assign/determine the fragment fold/structure?

Also at page 8, 1 line top – circular dichroism spectra are mentioned but not shown. This a pity as they represent the only experimental data of the secondary structure adopted by the A20 C-ter fragment. Also there is no mention of the methodology used to extract the 34% α -helical content.

The remaining section on the related polymerase structures and modeling of the E9/DNA complex mainly uses the SAXS technique to assess the conformational changes upon DNA binding and position/orientation of the thumb domain. However comparisons between model-calculated and experimental SAXS curves, although indicative – to a common reader – the sentence 'the fit is inferior as in the case' doesn't help much to grasp this 'inferiority'. Would it be possible to quantify with a 'number' this 'inferiority'?

Finally the Discussion section is interesting as it establishes structural relationships across different DNA polymerase species and their architecture with processivity factors with solid grounds for the

first and third paragraph. However the 'Modeling of the holoenzyme and processivity factor binding' paragraph relies on limited evidence.

In conclusion, it would seem that this manuscript would be more suitable for a more specialised journal or structural journal. Nevertheless, this is an editorial decision.

Minor points:

Page. 12 – Line 3 from top. How the $\pm 30\sigma$ accuracy has been estimated?

Page 12. – Lines 11 from top. Please change HCV for HSV

Page 28. – Table. The CC1/2 should also be included in the data collection statistics

Page 28. – Table. The Bond lengths 0.015 Å – is a bit too large, isn't it? Any reason for this?

Figure 1. Panel d and e – the ovals should be positioned in all four representations and possibly with white glowing around to make them stand out; e.g. in panel d left it's very hard to see the black oval.

Figure 6b. Possibly labeling of the 'finger' would help the reader.

Regarding the Figures – a stereoview of the electron density for the E9 catalytic domain should be provided.

Regarding the PDB report. The percentile scores for the Ramachandran and Sidechain outliers are a bit high relative to X-ray structure of similar resolution - but it seems that the possible deviations (poor fit in density) are located at the terminal ends possibly reflecting flexibility. What is the CC% of residue fit in density for these terminal residues?

Reviewer #3 (Remarks to the Author):

This paper by Tarbouriech et al, describes for the first time the structure of the catalytic polymerase component E9 of the Vaccinia virus polymerase complex, and as such will be of interest to both DNA virus virologists as well as those interested in the biochemistry of DNA replication. In addition they identify and purify a soluble domain of A20, which forms the heterodimeric processivity factor A20/D4 which binds E9. They show the A20 domain is ordered by SAXS and binds to E9, using both SEC and SPR. They identify a predicted C-terminal helix of A20 as a potential binding site for E9, and show mutation of a strictly conserved PHE is sufficient to break the interaction.

Using Mass spec and H2/D2 labelling they identify residues on E9 that bind to A20, and show convincingly they map to a helix in insert 3. Their structure of E9 is a holoenzyme and they perform SAXS experiments on E9 with and without DNA, which suggested domain movements in align with those seen in previous studies on family B DNA polymerase structures. My only major criticism of the paper is that there is a lot of text on explaining the various resistance mutations selected against several antiviral drugs, which is a bit "hand-wavy". Other than that the paper is concise, well written and clear, and provides the first insight into the molecular details of the E9/A20/D4 pol complex.

I have a number of specific points that the authors could consider addressing.

Page 3. "E9 alone was shown to be distributive..." what does this mean ??

Page 6. I assume the number is 4600. The number is split over 2 lines.

Page 8. The Hydrophobic residues. Can these be better labelled.....in figure 4 and also in the fig legend

Figure 1 Panel a needs to be bigger, showing the active residues, and could the orientation be more in the classic view, with the palm at the bottom. Also I don't understand why the calculated electrostatic potentials and conservation Fig1 d and e are drawn. What is the point of these ? I didn't see anything drawn out from these in the discussions ??

Figure 4. I thought could be moved to Supplementary. Also what are the mutants., what is it telling you. The reference in the text is poor. Perhaps move perhaps supp fig3 or supp fig 5 instead to the main text.

Figure 6. This may reflect my misreading of the paper, but would you not expect the movement of the finger towards the insert 2 domain, since you are superposing E9 on human pol alpha. The language is poor here. I think what the authors mean is that "Note the location of PAA resistance mutations, close to where the fingers domain moves on NTP binding". But this is all based on superposition on human pol alpha, which is fine, and mutations agree with the general model.

Table 1. The data go to higher resolution. This doesn't effect the results of the paper, but in their next publication the authors should process their data so that the CC1/2 is ~0.3 in the outer rs shell. They don't give CC1/2, but maybe they collected the data several years ago.

Response to reviewers

Reviewer #1 (Remarks to the Author):

The manuscript by Tarbouriech et al. presents the first crystal structure of the vaccinia virus DNA polymerase, and as such is a major advance in the field. For all viruses, the polymerase is the core of the replication machinery, a major determinant of genetic stability and variability, and a significant therapeutic target. As could have been predicted from sequence comparisons, the polymerase is a member of the B family of polymerases and its structure has the classic palm, thumb and finger components. Superimposed on this similarity, however, is the presence of several poxvirus-specific inserts that this same group noted in earlier publications. The current work provides new insight into the structure of these inserted sequences.

Another unique feature of the vaccinia polymerase is the nature of its processivity factor, a heterodimer of the A20 and D4 proteins. A20 binds to both the polymerase and D4, and serves as a bridge between these two proteins. Until now, there has been no insight into which region(s) of Pol bind to A20 and which region(s) of A20 bind to Pol. In this work, the authors determine that the C'-terminal 123 residues of A20 are sufficient to bind to Pol with a Kd of 23 nM. They also determine that the region of Pol that binds to A20 is the poxvirus-specific "insert 3" region. This finding is significant in that for many/most related polymerases, it is their C' terminus that binds to their processivity factor. For both A20 and Pol, residues that contribute to the interaction were identified using preliminary site-specific mutagenesis. Thus, this structural analysis of Pol has already enabled significant advances in the understanding of how the holoenzyme is assembled.

The work is well done, utilizes a number of different and effective techniques, and allows

important conclusions to be drawn. The work is important and will be of significant interest to both poxvirologists and those interested in polymerase structure/function. A few comments should be addressed:

- Czarnecki and Traktman (2017, *Virus Research*, PMID: 28159613) published a thorough review of the E9 DNA polymerase and the A20/D4 processivity factor earlier this year (before the current manuscript was submitted), with significant detail about genetic lesions and structural analysis. The authors should compare their mutant assignments and phenotype assignments with those in this recent review. The omission of this reference is a serious flaw that must be corrected.

The resistant mutants described in our study are in accordance with the ones presented in the review by Czarnecki and Traktman. However, we deliberately removed from our study some mutants (i.e., H296Y, R604S and H319N) because when introduced individually in a WT E9 background these mutations did not confer resistance against CDV (Kornbluth RS et al. 2006). Likewise, the mutation M671I could not be rescued on its own and always depended on a second contributing mutation for resistance (Becker M.N. et al. 2008).

We apologize that the reference to this recent review was lost when we shortened the introduction. It has been reintroduced now.

- The authors cite the work from the Evans laboratory showing that the vaccinia Pol is unusual in being able to mediate strand annealing reactions (in association with the single-strand binding protein I3). Does their structure provide any insight into this property?

No. From a structural point of view, no obvious domain could be related to recombinase activity. Furthermore, it has been shown that poxvirus DNA synthesis and recombination are tightly linked processes and to date genetic and biochemical studies failed to isolate E9 mutants with functionally separable activities. This is now briefly stated in the discussion.

- For a broader non-virological audience, it would be useful to have some general references for summary statements made in the Introduction about the sites(s) of viral DNA replication, etc.

The reference to a general review by Moss B. (2013) in *Fields Virology* is now added in the text.

- The list of proteins that are essential for viral replication is somewhat incomplete. For example, the viral DNA ligase (A50) appears to be vital for replication in the absence of the cellular ligase I.

The following sentence has been added in the introduction:

Other members of the virally encoded replication machinery include G5, a FEN-family endonuclease; A50, a DNA ligase and H5, an abundant hub protein¹⁰.

- On page 12, the abbreviation “HCV” is incorrect and should be changed to “HSV”, for herpes simplex virus (not hepatitis C virus).

HCV has been changed to HSV

- On page 6, while describing the preparation of a library of A20 fragments, the manuscript has the phrase “Approximately 4 600 clones were isolated and tested”. The meaning of this phrase is unclear and should be corrected.

The explanation of this step is now slightly more detailed. The paragraph was changed to:

We used the ESPRIT technology²⁹ to identify soluble purifiable C-terminal fragments of A20. A random library of 5' truncations of the full-length *A20R* gene was generated using exonuclease III and mung bean nuclease. Approximately 4 600 truncation mutants were isolated by robotic colony picking and tested for expression of soluble protein in an *E. coli* colony-based screen on nitrocellulose filters. Forty-eight clones were selected for scale-up and affinity purification testing.

NB: the reference describing ESPRIT was changed to a more recent one.

Reviewer #2 (Remarks to the Author):

The manuscript entitled ‘Structural basis of the novel mode of processivity factor binding by vaccinia virus DNA polymerase’ by Tarbouriech et al describes the X-ray structure of the catalytic subunit of the DNA polymerase E9 and locates on the palm domain a binding site for the C-terminal residues of the A20 processivity factor. From here the authors then initiate a comparative analysis with other DNA polymerase species and infer mechanisms of drug resistance.

General Comments

The article reads amenably. It’s nicely tailored as it combines complementary structural and biophysical techniques and it addresses an important area of research. However the breadth of the findings on which the discussion and conclusion are based, is questionable. The crystallographic study of the E9 DNA polymerase is solid although some minor revisions are required in the Table 1 (see below).

The section on the identification of a soluble A20 (49 kDa) domain interacting with E9 poses some questions. While the experimental part (ESPRIT technology) leading to this identification has proved a successful strategy to recognize soluble portions of a protein, the major concern is that the found A20 C-ter might not be the only region interacting with the E9 polymerase. The evidence on which the interaction between E9 and the C-terminal domain of A20 is hypothesized comes from SAXS studies on the D4-A20-E9 complex that provided an envelope in which the known/available structures at the time were manually fitted (see Fig. 8 in Ref 9) – there are no other biochemical or cross-link mass-spec exps on the E9 and full-length A20 complex, for example, that would further strengthen/corroborate this and map possible interacting residues between the two subunits, including those assumed with the A20 C-ter. And while it has been proved that the A20 C-ter and E9 interacts via gel chromatography and SPR (and further in the text identified the putative interacting residues via LC-MS/MS) – data supporting the exclusiveness of this C-ter of the A20 region are limited, a fact that undermines the generality of the findings of this aspect of the work.

To answer to the comments concerning the putative domains of A20 interacting with E9 we performed a new experiment that is now presented in Supplementary Fig 3 f,g.

In this experiment, the Phe414Ala mutation was introduced into the full-length VACV A20 protein. We then purified the heterodimeric A20/D4 complexes (WT and mutant) expressed in insect cells and checked their ability to bind to VACV E9. We showed using SPR that while WT A20/D4 interacts with E9, binding of the mutant complex was strongly reduced. The dramatic loss of A20/E9 interaction upon a single mutation in the putative C-ter α -helix of A20 further strengthens the model that the C-terminal domain of A20 harbors the main E9 binding site.

Moreover, using SAXS ab initio on a 123 residues protein fragment (A20 C-ter) might be stretching the limit of envelope interpretation (as also the fitting of a SAXS envelope in another previously determined SAXS envelope, Fig 5a). Regarding this (A20 C-ter) it would have been useful to explain in the SI what restraints were used that led to the final 'best' model envelope (Suppl. Fig 2), and of course to show the second and third best solution.

The fit of the A20 C-ter envelope into the envelope of the E9/A20/D4 complex is only manual and obviously quite approximate. The consensus on the shape of the bead models for A20 C-ter on the other hand is particularly good. We join a figure for the referees (see Fig. 1 below), which shows also the 2nd and 3rd best solutions as well as the consensus from 40 models.

As for the identification of the interacting residues the described logic is solid and linear with the combination of mutagenesis and mass-spec techniques and yet the entire framework is built on a secondary structure prediction of the last 27 residues of the A20 C-ter being α -helical (BTW there is no confidence scale or reliability score in panel c of Fig. 2).

A confidence score has been added in Fig. 2c. Due to the unavailability of the jpred server, an equivalent prediction with MLRC from the NPSA server has been used and the bibliographic reference has been changed accordingly.

As this is an important assumption, much stronger and further experimental data would have been beneficial to prove the case. For example, why not synthesise the last 30 residues and check, for example, by CD the helical propensity of the simple fragment, or as the A20- C-ter is produced in bacteria why not collaborate with a NMR group assign/determine the fragment fold/structure?

A peptide corresponding to the putative C-ter α -helix of A20 was purchased for attempts to co-crystallize E9 with this fragment. Unfortunately, the peptide could only be solubilized in 100% DMSO.

A collaboration with a NMR group on the structure of A20 C-ter is planned in the future.

Also at page 8, 1 line top – circular dichroism spectra are mentioned but not shown. This a pity as they represent the only experimental data of the secondary structure adopted by the A20 C-ter fragment. Also there is no mention of the methodology used to extract the 34% α -helical content.

The CD spectrum of A20 C-ter is now presented in the paper in Supplementary Figure 2 d. In order to improve the quality of the spectra, A20 C-ter WT and mutant were dialyzed in phosphate buffer and sodium fluoride (NaF) as stated in the methods section. A helical content of 45 % was obtained for the WT protein using Contin/Dichroweb. The analysis of the Phe414Ala mutant gives similar results (43%).

The remaining section on the related polymerase structures and modeling of the E9/DNA complex mainly uses the SAXS technique to assess the conformational changes upon DNA binding and position/orientation of the thumb domain. However comparisons between model-calculated and experimental SAXS curves, although indicative – to a common reader - the sentence ‘the fit is inferior as in the case’ doesn’t help much to grasp this ‘inferiority’. Would it be possible to quantify with a ‘number’ this ‘inferiority’?

As structural differences will only show up in certain ranges of the value of the scattering vectors, the visual comparison is a valid criteria in order to judge the quality of the fit. Fitness parameters for the comparison of the model and the full scattering curve such as the chi2 could be calculated but their reliability is limited. Using the data set of the complex, the chi2 are 8.6 for the apo form 1.6 for the model where only the thumb position has been adjusted and 1.1 for the domain-wise fit. These values are now given in Suppl. Fig. 5. The sentence ‘the fit is inferior as in the case’ has been modified.

Finally the Discussion section is interesting as it establishes structural relationships across different DNA polymerase species and their architecture with processivity factors with solid grounds for the first and third paragraph. However the ‘Modeling of the holoenzyme and processivity factor binding’ paragraph relies on limited evidence.

Additional evidence for the different interaction mode is now given in the section: As electron density for all the C-terminal residues is visible an involvement of the C-terminus in interactions with processivity factors as observed for other polymerases is formally excluded. On the other hand we state now that additional interactions with other domains cannot be fully excluded.

In conclusion, it would seem that this manuscript would be more suitable for a more specialised journal or structural journal. Nevertheless, this is an editorial decision.

Minor points:

Page. 12 – Line 3 from top. How the $\pm 30^\circ$ accuracy has been estimated?

The corresponding sentence has been changed in order to clarify the approach.

We repeated the modelling and prefer to give an accuracy of $\pm 40^\circ$ now. The model has been docked manually into the SAXS envelope of the holoenzyme (Sèle et al., 2013) and an intermediate orientation of E9 is shown. An angle of 80° has been measured between the 2 extreme orientations of E9 within the envelope which are compatible with a contact of the helix of insert 3 with A20 when A20 is also positioned inside the envelope as described by Sèle et al. 2013.

Page 12. – Lines 11 from top. Please change HCV for HSV

This has been done

Page 28. – Table. The CC1/2 should also be included in the data collection statistics

The CC1/2 have been included in Table 1.

Page 28. – Table. The Bond lengths 0.015 Å – is a bit too large, isn't it? Any reason for this?

REFMAC with its default parameters has been used for the refinement. Compared with Engh-Huber standard bond lengths (Engh and Huber 1991) using PROCHECK (Laskowski et al. 1993), the rms value of 0.015 appears to be still very tight. The output from PROCHECK is provided for the reviewers (see Fig. 2 below).

Figure 1. Panel d and e – the ovals should be positioned in all four representations and possibly with white glowing around to make them stand out; e.g. in panel d left it's very hard to see the black oval.

This has been done

Figure 6b. Possibly labeling of the 'finger' would help the reader.

This has been done. For clarity the panel has also been simplified.

Regarding the Figures – a stereoview of the electron density for the E9 catalytic domain should be provided.

We feel that it is sufficient to include a stereo view of the electron density in the Supplementary Material. Supplementary figure 4 has now been changed in order to include stereo views and electron density of the exonuclease active site for the WT E9 structure and the $\text{exo}^{\text{minus}}$ mutant in panels a and b.

Regarding the PDB report. The percentile scores for the Ramachandran and Sidechain outliers are a bit high relative to X-ray structure of similar resolution - but it seems that the possible deviations (poor fit in density) are located at the terminal ends possibly reflecting flexibility. What is the CC% of residue fit in density for these terminal residues?

The high temperature factor in the thumb domain and insert 2 combined with a section of residues 526-531 at the beginning of the palm domain which shows a possible alternate conformation limit the quality of the model. Still the trace of the peptide chain globally is well defined. The additional residues from cloning are visible.

A detailed output from SFCHECK is joined (in a separate file).

Reviewer #3 (Remarks to the Author):

This paper by Tarbouriech et al, describes for the first time the structure of the catalytic polymerase component E9 of the Vaccinia virus polymerase complex, and as such will be of interest to both DNA virus virologists as well as those interested in the biochemistry of DNA replication. In addition they identify and purify a soluble domain of A20, which forms the heterodimeric processivity factor A20/D4 which binds E9. They show the A20 domain is ordered by SAXS and binds to E9, using both SEC and SPR. They identify a predicted C-terminal helix of A20 as a potential binding site for E9, and show mutation of a strictly conserved PHE is sufficient to break the interaction.

Using Mass spec and H2/D2 labelling they identify residues on E9 that bind to A20, and show convincingly they map to a helix in insert 3. Their structure of E9 is a holoenzyme and they perform SAXS experiments on E9 with and without DNA, which suggested domain movements in align with those seen in previous studies on family B DNA polymerase structures. My only major criticism of the paper is that there is a lot of text on explaining the various resistance mutations selected against several antiviral drugs, which is a bit “hand-wavy”. Other than that the paper is concise, well written and clear, and provides the first insight into the molecular details of the E9/A20/D4 pol complex.

A precise explanation of the action of resistance mutants will require high-resolution studies of the relevant DNA bound states for the WT and the mutant, which is far beyond the scope of the paper.

I have a number of specific points that the authors could consider addressing.

Page 3. “E9 alone was shown to be distributive...” what does this mean??

The term “distributive” is now better explained: The sentence is now as follows:

E9 alone was shown to be distributive under physiological conditions (adding fewer than 10 nucleotides per binding event) unless bound to its heterodimeric cofactor D4/A20.

Page 6. I assume the number is 4600. The number is split over 2 lines.

This is now corrected

Page 8. The Hydrophobic residues. Can these be better labelled.....in figure 4 and also in the fig legend

They are now explicitly listed in the text and a new color scheme is used in Fig. 4.

Figure 1 Panel a needs to be bigger, showing the active residues,

At that stage, there is no detailed discussion of individual residues which would need localization on the structure. So we feel that the views presented are detailed enough. According to a request of reviewer 2, we included a stereo view of the exonuclease site together with its electron density in Supplementary Figure 4.

and could the orientation be more in the classic view, with the palm at the bottom.

An analysis of several publications on family B polymerases listed below showed that there is no preferred orientation in the presentation of the polymerase.

Doublié, Sylvie, and Karl E. Zahn. 2014. “Structural Insights into Eukaryotic DNA Replication.” *Frontiers in Microbiology* 5: 444. doi:10.3389/fmicb.2014.00444.
Liu, S., J. D. Knafels, J. S. Chang, G. A. Waszak, E. T. Baldwin, M. R. Deibel, D. R. Thomsen, et al. 2006. “Crystal Structure of the Herpes Simplex Virus 1 DNA Polymerase.” *J Biol Chem* 281 (June): 18193–200.
Swan, Michael K., Robert E. Johnson, Louise Prakash, Satya Prakash, and Aneel K. Aggarwal. 2009. “Structural Basis of High-Fidelity DNA Synthesis by Yeast DNA

Polymerase Delta.” Nature Structural & Molecular Biology 16 (9): 979–86.
doi:10.1038/nsmb.1663.

Wang, Feng, and Wei Yang. 2009. “Structural Insight into Translesion Synthesis by DNA Pol II.” Cell 139 (7): 1279–89. doi:10.1016/j.cell.2009.11.043.

Wang, J., A. K. Sattar, C. C. Wang, J. D. Karam, W. H. Konigsberg, and T. A. Steitz. 1997. “Crystal Structure of a Pol Alpha Family Replication DNA Polymerase from Bacteriophage RB69.” Cell 89 (7): 1087–99.

Also I don't understand why the calculated electrostatic potentials and conservation Fig 1 d and e are drawn. What is the point of these? I didn't see anything drawn out from these in the discussions??

The sequence conservation (Fig. 1e) is used as it points to a role of insert 3.

A sentence exploiting sequence conservation and electrostatics for insert 2 has been added in the discussion:

“It only shows weak sequence conservation between poxviruses (Fig. 1e) and the absence of strong electrostatic features (Fig. 1d) make it unlikely that this domain interacts with nucleotides. Interestingly, insert 2 carries a number of resistance mutations towards PAA (Table 2) that could reveal a possible contact between this domain and the finger domain in the presence of an incoming nucleotide (discussed below).”

Figure 4. I thought could be moved to Supplementary. Also what are the mutants, what is it telling you. The reference in the text is poor. Perhaps move perhaps supp fig3 or supp fig 5 instead to the main text.

Conserving the number of display items in the article we moved Supplementary Fig. 3 into the main text and Fig. 3 into the supplementary material. The new supplementary Fig. 3 has been enriched by 2 panels on the binding of D4/A20 and its Phe414Ala mutant to E9 measured by SPR. We still think that Fig. 4 is central for the article and leave it in the current position, although the color scheme has been changed for easier understanding.

Figure 6. This may reflect my misreading of the paper, but would you not expect the movement of the finger towards the insert 2 domain, since you are superposing E9 on human pol alpha. The language is poor here. I think what the authors mean is that “Note the location of PAA resistance mutations, close to where the fingers domain moves on NTP binding”. But this is all based on superposition on human pol alpha, which is fine, and mutations agree with the general model.

The sentence has been changed in an attempt of clarification.

“Upon nucleotide binding the tip of the finger of E9 may move close to the insert 2 domain allowing a potential contact, which would occur in vicinity of the PAA resistance mutations depicted as green spheres.”

Table 1. The data go to higher resolution. This doesn't affect the results of the paper, but in their next publication the authors should process their data so that the CC1/2 is ~0.3 in the outer rs shell. They don't give CC1/2, but maybe they collected the data several years ago.

Indeed we stay quite conservative in the resolution cut-off as it was current practice several years ago. On the other hand the total intensity and information content of the omitted reflections is extremely low, so that there is no practical effect on the quality of the model. Data were actually collected between February 2014 and January 2016. A line with the CC1/2 values is now included in Table 1.

Fig 1: Comparison of the different *ab initio* bead models obtained from 40 runs of DAMMIF:

Best

2nd

3rd

Consensus

Fig 2: Procheck analysis of the E9 structure regarding bond lengths:

REVIEWERS' COMMENTS:

Reviewer #2 (Remarks to the Author):

The authors have addressed most of the raised concerns.
However four points need to be addressed before final publication.

First, for the CD spectrum the authors should provide the reconstructed curve and the experimental one superimposed and the normalized root-mean-square deviation (NRMSD).

Second, the Fig. 1 (Comparison of the different ab initio beads models obtained from 40 runs of DAMMIF) provided to the referee only should be included as a further panel in the Supplementary Figure 2. The relevance to have it included in the manuscript is to provide the reader less acquainted with the SAXS technique with a 'visual criteria' for the choice of the best model over the 2nd and 3rd best ones.

Third, the section 'Identification of residues involved in A20/E9 interaction' from page 7 (line 144) to page 9 (line 199) requires some additional ordering of the concepts presented. For example: at page 7 line 142 the reader is referred to Supplementary Fig. 3a then at page 8 line 164 the reader is referred to Supplementary Fig. 3e – so where are the Supplementary Figs 3b-d ? these are referred to in the paragraph at page 9 from line 192 to line 196 and after the Supplementary Figs. 3f and 3g have been presented (page 8 line 168 and line 171 respectively).

Forth, page 13 from line 266 to line 274: Fig. 5b is not mentioned between Fig. 5a and Figure 5c,d,e – only at page 14 line 292. Can be the ordering of panels or concepts presented revised accordingly?

General comment on the X-ray structure:

The PDB validation report and SFCHECK seems to show that the refined structure doesn't score particularly high in terms of percentiles in both protein backbone and sidechains geometry and with quite a bit of residues with a $B > 60 \text{ \AA}^2$. However other indicators suggest that this might be due to the inherent flexibility of the molecule rather than to the strategy of the refinement adopted (E9 is a relatively large molecule and the resolution achieved is not 'high' - 2.7-2.8 Å).

At page 19 line 411 the authors simply state that refinement was performed with REFMAC – possibly a couple of more details on the strategy would benefit the general reader (a more curious one can go and look at the corresponding reference). For example –was individual B-factor refinement used? Have been TLS used? If so, how many? This information can be added easily into the text.

Minor points:

Page 6 – line 115: 4600 (there is a space too much);

Page 7 – line 129: α -helix;

Page 10 – line 228: please use 'supporting' rather than 'confirming'.

Response to referee

Reviewer #2 (Remarks to the Author):

The authors have addressed most of the raised concerns.
However four points need to be addressed before final publication.

First, for the CD spectrum the authors should provide the reconstructed curve and the experimental one superimposed and the normalized root-mean-square deviation (NRMSD).

Supplementary fig 2d was modified. The reconstructed curve and the experimental one are now superimposed. The normalized root-mean-square deviation (NRMSD) is indicated in panel d.

Second, the Fig. 1 (Comparison of the different ab initio beads models obtained from 40 runs of DAMMIF) provided to the referee only should be included as a further panel in the Supplementary Figure 2. The relevance to have it included in the manuscript is to provide the reader less acquainted with the SAXS technique with a 'visual criteria' for the choice of the best model over the 2nd and 3rd best ones.

2nd, 3rd and consensus SAXS models were included in Supplementary fig 2c as proposed by reviewer.

Third, the section 'Identification of residues involved in A20/E9 interaction' from page 7 (line 144) to page 9 (line 199) requires some additional ordering of the concepts presented. For example: at page 7 line 142 the reader is referred to Supplementary Fig. 3a then at page 8 line 164 the reader is referred to Supplementary Fig. 3e –

so where are the Supplementary Figs 3b-d ? these are referred to in the paragraph at page 9 from line 192 to line 196 and after the Supplementary Figs. 3f and 3g have been presented (page 8 line 168 and line 171 respectively).

The panel order from supplementary Fig.3 was changed as well as Figure citation in the text. Panels from this Fig are now cited chronologically. The figure legend was modified accordingly.

Forth, page 13 from line 266 to line 274: Fig. 5b is not mentioned between Fig. 5a and Figure 5c,d,e – only at page 14 line 292. Can be the ordering of panels or concepts presented revised accordingly?

Fig 5b is now cited together with Fig. 5c,d,e.

General comment on the X-ray structure:

The PDB validation report and SFCHECK seems to show that the refined structure doesn't score particularly high in terms of percentiles in both protein backbone and sidechains geometry and with quite a bit of residues with a $B > 60 \text{ \AA}^2$. However other indicators suggest that this might be due to the inherent flexibility of the molecule rather than to the strategy of the refinement adopted (E9 is a relatively large molecule and the resolution achieved is not 'high' - 2.7-2.8 \AA).

At page 19 line 411 the authors simply state that refinement was performed with REFMAC – possibly a couple of more details on the strategy would benefit the general reader (a more curious one can go and look at the corresponding reference). For example –was individual B-factor refinement used? Have been TLS used? If so, how many? This information can be added easily into the text.

The required information was added to the text : TLS were not used and only individual isotropic B factors were used. We agree that the refinement indicators are not better due to the inherent flexibility of the protein and high solvent content of the crystals.

Minor points:

Page 6 – line 115: 4600 (there is a space too much); **Done**

Page 7 – line 129: α -helix; **Done**

Page 10 – line 228: please use ‘supporting’ rather than ‘confirming’. **Done**